# The Entropic Signature of Class Speciation in Diffusion Models

Florian Handke [* 1]  Dejan Stančević [* 2]  Felix Koulischer [1]  Thomas Demeester [1]  Luca Ambrogioni [2]

## Abstract

Diffusion models do not recover semantic structure uniformly over time. Instead, samples transition from semantic ambiguity to class commitment within a narrow regime. Recent theoretical work attributes this transition to dynamical instabilities along class-separating directions, but practical methods to detect and exploit these windows in trained models are still limited. We show that tracking the class-conditional entropy of a latent semantic variable given the noisy state provides a reliable signature of these transition regimes. By restricting the entropy to semantic partitions, the entropy can furthermore resolve semantic decisions at different levels of abstraction. We validate our method on EDM2-XS and Stable Diffusion 1.5, where class-conditional entropy consistently isolates the noise regimes critical for semantic structure formation. Finally, we use our framework to quantify how guidance redistributes semantic information over time. Together, these results connect information-theoretic and statistical physics perspectives on diffusion and provide a principled basis for time-localized control.

## 1. Introduction

Diffusion models have become state-of-the-art generative frameworks for images, audio, video, and multimodal settings (Sohl-Dickstein et al., 2015; Ho et al., 2020; Song et al., 2021b; Kong et al., 2021; Singer et al., 2023). In modern applications, their practical utility is closely linked to their ability to follow conditioning signals (labels, text, or other modalities) by guiding the reverse-time denoising dynamics toward user-specified semantics (Dhariwal &

Nichol, 2021; Ho & Salimans, 2022). Guidance, especially classifier-free guidance, underlies many of the most capable conditional generators, including GLIDE, Imagen, and DALL·E-style systems, as well as latent-diffusion-based models such as Stable Diffusion (Nichol et al., 2022; Saharia et al., 2022; Ramesh et al., 2022; Betker et al., 2023; Rombach et al., 2022). Despite the strong empirical performance of guided diffusion, the mechanisms by which semantic structure emerges during the denoising process, and how this emergence relates to actionable control in sampling, remain only partially understood.

A growing body of theoretical literature connects sampling in diffusion models with ideas from statistical physics, including dynamical instabilities, phase transitions, and symmetry breaking in high-dimensional stochastic flows (Biroli & Mézard, 2023; Biroli et al., 2024; Montanari, 2023; Montanari & Wu, 2023; Ambrogioni, 2025). A prominent line of work studies the reverse-time dynamics under exact or near-exact score assumptions and identifies a sharp transition from semantic ambiguity to class commitment, termed symmetry-breaking class speciation (Raya & Ambrogioni, 2023; Biroli et al., 2024). In this picture, speciation corresponds to a time when the coarse structure of the data is revealed. Complementary studies show that guidance interacts non-trivially with these dynamical regimes, particularly in high dimensions, helping to explain why classifier-free guidance is effective in practice despite known low-dimensional pathologies (Pavasovic et al., 2025). In parallel, empirical and theoretical results report the emergence of phase-transition-like features and narrow critical windows along the sampling trajectory, suggesting that different semantic attributes are determined at different noise scales (Sclocchi et al., 2025; Li & Chen, 2024). However, existing theories do not provide an operational quantity that tracks semantic commitment along the sampling trajectory, aligns with the symmetry-breaking/speciation picture, and can be directly estimated in trained diffusion models.

Our contributions are as follows: (i) theoretically, we show that class-conditional entropy provides an operational marker of speciation, detecting not only the transition in high-dimensional Gaussian mixtures but also the more general speciation times introduced by Achilli et al. (2026); (ii) we demonstrate that this signal can be estimated in trained diffusion models and allows isolating the emergence of spe-

*Equal contribution  [1]Department of Information Technology, Ghent University - imec, Ghent, Belgium  [2]Donders Institute for Brain, Cognition and Behaviour, Radboud University, Nijmegen, the Netherlands. Correspondence to: Florian Handke <florian.handke@ugent.be>, Dejan Stancevic <dejan.stancevic@donders.ru.nl>.

*Proceedings of the 43rd International Conference on Machine Learning*, Seoul, South Korea. PMLR 306, 2026. Copyright 2026 by the author(s).

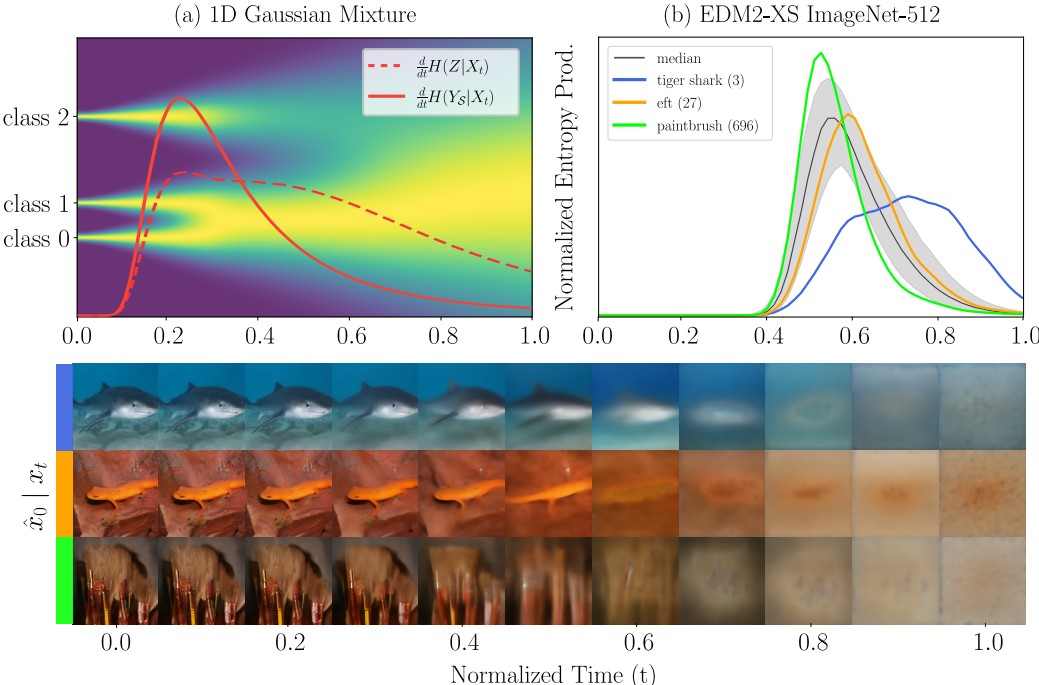

*Figure 1.* **Overview of entropy production in generative diffusion**. (a) Entropy production quantified as the temporal derivative of the (partitioned) conditional entropy during the denoising process. The dashed curve shows the class-conditional entropy production over the full label space, capturing semantic commitment at the level of the complete class variable. The solid curve shows the partitioned class-conditional entropy production for the binary decision between class 0 and class 1, isolating when this specific semantic distinction is resolved. (b) Entropy production in a trained model. Each curve shows the partitioned class-conditional entropy production for a single class against its complement. The lower panel displays representative noiseless predictions along the denoising trajectory for the highlighted classes, illustrating how semantic structure emerges around the corresponding entropy production peaks.

cific semantic elements in practice; and (iii) we use it to analyze how guidance redistributes semantic information over time.

We validate our theory on synthetic Gaussian mixtures and demonstrate its practical relevance on EDM2 (Karras et al., 2024) and Stable Diffusion (Rombach et al., 2022), trained on ImageNet (Deng et al., 2009) and LAION5B (Schuhmann et al., 2022), respectively. Figure 1 illustrates the resulting entropy-production picture, i.e., the temporal derivative of the class-conditional entropy, in both a simple one-dimensional setting and a realistic dataset. In all settings, class-conditional entropy consistently identifies the noise regimes in which class-specific features emerge and quantifies the distortion of information recovery induced by guidance, thus complementing recent empirical findings on its limited effectiveness in low- and high-noise regimes (Kynkäänniemi et al., 2024; Koulischer et al., 2025b).

## 2. Related Work

**Information theory**    In recent years, the connection between diffusion models and information theory has been studied in more detail. Kong et al. (2023) explore the information-theoretic underpinnings of diffusion models

and provide practical estimators for likelihood-related quantities, while Franzese et al. (2024) show how score-based diffusion models can be used to estimate mutual information (and related entropic quantities). Premkumar (2025) studies properties of a trajectory-level entropy motivated by stochastic-thermodynamic viewpoints. In later work, Premkumar (2026) draws an analogy between diffusion models and the Kelly criterion, giving a direct relation to the amount of additional information required to store class-conditional structure. Stancevic et al. (2026), on the other hand, focus on using an entropy-based scheduler to accelerate sampling in such a way that each step contributes a comparable information gain to the final sample. Stančević & Ambrogioni (2026) further develop this viewpoint and connects it to the physics picture of diffusion, establishing entropy and information flow as central quantities for analyzing diffusion dynamics.

While Wang et al. (2025) use mutual information between the text prompt and the final generated image as an explicit alignment objective and diagnostic for prompt-conditioned generation, Kong et al. (2024) treat mutual information and related information-decomposition quantities as interpretability signals that attribute how different factors contribute to the final sample. In both cases, mutual information

summarizes how strongly the conditioning affects the output. By contrast, we treat semantic commitment as a dynamical process. We track class-conditional uncertainty along the denoising trajectory. Also, we show theoretically that the resulting transition region matches the speciation window predicted by high-dimensional analysis.

**Physics perspective** Several works characterize semantic emergence in diffusion sampling as a sharp high-dimensional transition, including the speciation instabilities (Biroli et al., 2024; Achilli et al., 2026) and related symmetry-breaking and critical-window perspectives (Raya & Ambrogioni, 2023; Ambrogioni, 2025; Sclocchi et al., 2025; Li & Chen, 2024). These analyses provide mechanistic explanations but do not, in general, yield a simple online diagnostic for trained models or a direct prescription for time-localized intervention. Two notable steps toward practice are the work from Pavasovic et al. (2025), which leverages the high-dimensional regime picture to explain and improve classifier-free guidance, and from Kynkäänniemi et al. (2024), which empirically shows that guidance is most beneficial when applied in a limited time interval. In contrast, we show that the class-conditional entropy detects not only the speciation transition identified by Biroli et al. (2024), but also the more general speciation times introduced by Achilli et al. (2026), including cases where classes are not separated by first moments alone. Thus, it provides a direct and operational signature of symmetry breaking in diffusion dynamics. Moreover, because it can be estimated along the denoising trajectory, it offers a practical diagnostic for guided diffusion.

## 3. Preliminaries

### 3.1. Diffusion Models

Diffusion models view data generation as a sequential stochastic transformation between two probability distributions: a complex data distribution and a simple noise distribution. This transformation is defined by stochastic differential equations (SDEs). The forward SDE gradually destroys information about the data by injecting noise, while the reverse SDE restores structure by progressively removing noise. Formally, an SDE specifies how a random variable evolves with time:

$$\mathrm{d}X_t = f(X_t, t)\,\mathrm{d}t + g_t\,\mathrm{d}W_t, \tag{1}$$

where $W_t$ is a standard Wiener process, $f$ is a drift field, and $g$ is a time-dependent diffusion coefficient controlling the variance of the injected noise. The evolution of marginal densities $p_t(x)$ is governed by the associated Fokker–Planck equation:

$$\partial_t p_t(x) = \sum_{j=1}^{d} \partial_{x_j} \left( -f_j(x, t) + \frac{g_t^2}{2}\partial_{x_j} \right) p_t(x). \tag{2}$$

For many choices of $(f, g)$, the forward process admits a closed-form transition kernel $p(x_t|x_0)$, allowing us to efficiently sample noisy states without explicitly simulating the SDE. Examples of widely used SDEs are Variance-Preserving (VP) (Song et al., 2021b) and Elucidated Diffusion Models (EDM) (Karras et al., 2022).

The generative model is obtained by reversing the forward process. Under mild conditions, Anderson (1982) showed that reversing an SDE introduces an additional drift term involving the score $\nabla \log p_t(x)$:

$$\mathrm{d}X_t = \left( f(X_t, t) - g_t^2 \nabla_x \log p_t(X_t) \right) \mathrm{d}t + g_t\,\mathrm{d}\widetilde{W}_t. \tag{3}$$

If the score were known exactly, sampling this reverse SDE (initialized from the approximately Gaussian final state of the forward process) would yield exact data samples. In practice, a neural network $s_\theta(x, t)$ is trained to approximate the score, which can then be used to numerically solve the reverse SDE to synthesize data.

To generate samples consistent with some label or prompt $y$, one replaces the unconditional score with the conditional score $\nabla_x \log p_t(x|y)$. In practice, techniques such as classifier guidance and classifier-free guidance are used (Ho & Salimans, 2022; Dhariwal & Nichol, 2021).

### 3.2. Information Theory

Information theory provides a language for quantifying uncertainty and information flow in stochastic processes. In the context of diffusion models, the forward SDE can be interpreted as transmitting a data point through a progressively noisier channel, whereas the reverse SDE attempts to infer the original symbol from its corrupted versions. This perspective makes information-theoretic quantities natural tools for understanding how much of the original structure remains at different times $t$. The fundamental measure of uncertainty is the (differential) entropy:

$$h(X) = -\int p(x) \log p(x)\,\mathrm{d}x, \tag{4}$$

which has a discrete counterpart, the *Shannon* entropy, which we denote using $H$. As we are interested in how much information about a latent variable is preserved as the diffusion process evolves, a central object is the *conditional entropy*:

$$h(Y|X) = -\iint p(x, y) \log p(y|x)\,\mathrm{d}x\,\mathrm{d}y, \tag{5}$$

which in our case becomes a mixed entropy (discrete in the latent variable, continuous in the state random variable). The conditional entropy quantifies the remaining uncertainty about $Y$ given knowledge of $X$. When applied to diffusion trajectories, it captures how uncertainty about the underlying data or conditioning variable evolves as noise is added or removed.

# 4. Theoretical Analysis

In this section, we introduce our theoretical setup, provide an exact expression for the class-conditional entropy production, and relate its peaks to speciation transitions. This identifies class-conditional entropy as a detector of speciation times. We make the mechanism explicit for Gaussian mixtures and extend it to a more general class-structure framework presented in Achilli et al. (2026). Details and derivations are provided in Appendix A.

## 4.1. Problem Setup: Semantic Uncertainty

We consider a general diffusion forward process for which the transition kernel is Gaussian and isotropic:

$$p(x_t|x_0) = \mathcal{N}\big(x_t|\alpha_t x_0, \sigma_t^2 I_d\big), \tag{6}$$

with scalar schedules $\alpha_t \geq 0$ and $\sigma_t \geq 0$. This form covers commonly used SDE parameterizations (such as VP and VE/EDM).

Let $Z \in \{1, \ldots, K\}$ denote a latent semantic variable (e.g., a class label) with prior $p(Z = k)$, and let $X_t$ be the noisy state random variable at time $t$. Our goal is to track when semantics become ambiguous under the forward process (and, equivalently, when they become resolved along the reverse process). This can be quantified by the class-conditional entropy

$$H(Z|X_t) = -\int p_t(x) \sum_{k=1}^{K} \gamma_k(x,t) \log \gamma_k(x,t) \, dx. \tag{7}$$

where $\gamma_k(x,t) := p_t(Z = k|x)$. Differentiating in $t$ yields an expression for the instantaneous rate of information transfer, which can be written as:

$$\frac{\partial}{\partial t} H(Z|X_t) = \frac{g_t^2}{4} \sum_{r,s=1}^{K} p(Z = r) \, \mathcal{E}_{r,s}(t), \tag{8}$$

where

$$\mathcal{E}_{r,s}(t) := \mathbb{E}\big[\gamma_s(X_t,t) \|s_r(X_t,t) - s_s(X_t,t)\|_2^2 \mid Z = r\big]. \tag{9}$$

This object depends on how much posterior mass is assigned to class $s$ and how different the two score fields are. A useful quantity for tracking this pairwise competition is the log-posterior ratio

$$\Lambda_{rs}(x,t) := \log \frac{\gamma_r(x,t)}{\gamma_s(x,t)}. \tag{10}$$

This quantity measures how strongly $x_t$ favors class $r$ over class $s$. Values near zero correspond to strong mixing between the two classes, while large absolute values indicate that they are well separated at noise level $t$. In the following, we show how these pairwise log-odds govern the entropy dynamics.

## 4.2. Gaussian Mixtures

To make the mechanism above explicit, we instantiate it for an equiprobable isotropic Gaussian mixture:

$$p_0(x) = \frac{1}{K} \sum_{k=1}^{K} \mathcal{N}\big(x|\mu_k, \sigma_0^2 I_d\big), \tag{11}$$

with component means satisfying the high-dimensional scaling $\frac{\|\mu_k\|_2^2}{d} = q_k$, $\frac{\|\mu_k - \mu_i\|_2^2}{d} = \delta_{ik}^2$ for $i \neq k$, where $q_k$'s and $\delta_{ik}^2$'s are positive constants as $d \to \infty$. Under the Gaussian kernel (6), each component remains Gaussian:

$$X_t \mid (Z = k) \sim \mathcal{N}\big(m_k(t), v(t) I_d\big), \tag{12}$$

with $m_k(t) = \alpha_t \mu_k$ and $v(t) = \alpha_t^2 \sigma_0^2 + \sigma_t^2$. In this model, both ingredients in (9) can be written in a closed form. The pairwise log-ratio (10) becomes:

$$\Lambda_{ik}(x,t) = \frac{\|m_k(t)\|_2^2 - \|m_i(t)\|_2^2 + 2x^\top\big(m_i(t) - m_k(t)\big)}{2v(t)}. \tag{13}$$

Conditioning on class $Z = i$ and writing $X_t = m_i(t) + \sqrt{v(t)}\,\varepsilon$ with $\varepsilon \sim \mathcal{N}(0, I_d)$ gives:

$$\Lambda_{ik}(X_t, t) = \frac{\|\Delta_{ik}(t)\|_2^2}{2v(t)} + \frac{\Delta_{ik}(t)^\top \varepsilon}{\sqrt{v(t)}}, \tag{14}$$

where $\Delta_{ik}(t) := m_i(t) - m_k(t)$. Defining the pairwise control parameter as:

$$\lambda_{ik}(t) := \frac{\|\Delta_{ik}(t)\|_2^2}{v(t)}, \tag{15}$$

we can see that, given samples from class $i$, $\Lambda_{ik}(X_t, t)$ is Gaussian with mean and variance respectively:

$$\mathbb{E}[\Lambda_{ik}(X_t, t) \mid Z = i] = \frac{\lambda_{ik}(t)}{2}, \tag{16}$$

$$\mathrm{Var}[\Lambda_{ik}(X_t, t) \mid Z = i] = \lambda_{ik}(t). \tag{17}$$

Since the Gaussian component scores are linear, we obtain for the score gap:

$$\|s_i(x,t) - s_k(x,t)\|_2^2 = \frac{\|\Delta_{ik}(t)\|_2^2}{v(t)^2} = \frac{\lambda_{ik}(t)}{v(t)}. \tag{18}$$

The same parameter $\lambda_{ik}(t)$ governs both posterior competition and score separation. This immediately suggests three regimes. If $\lambda_{ik}(t) \gg 1$, then the deterministic part of $\Lambda_{ik}$ dominates its fluctuations, so the posterior weight of the competing class is exponentially suppressed. Although the score gap is still large in that regime, it grows only linearly in $\lambda_{ik}(t)$, whereas the posterior factor decays exponentially in $\lambda_{ik}$. Hence the exponential suppression wins and the pairwise contribution remains negligible. If $\lambda_{ik}(t) \ll 1$, the

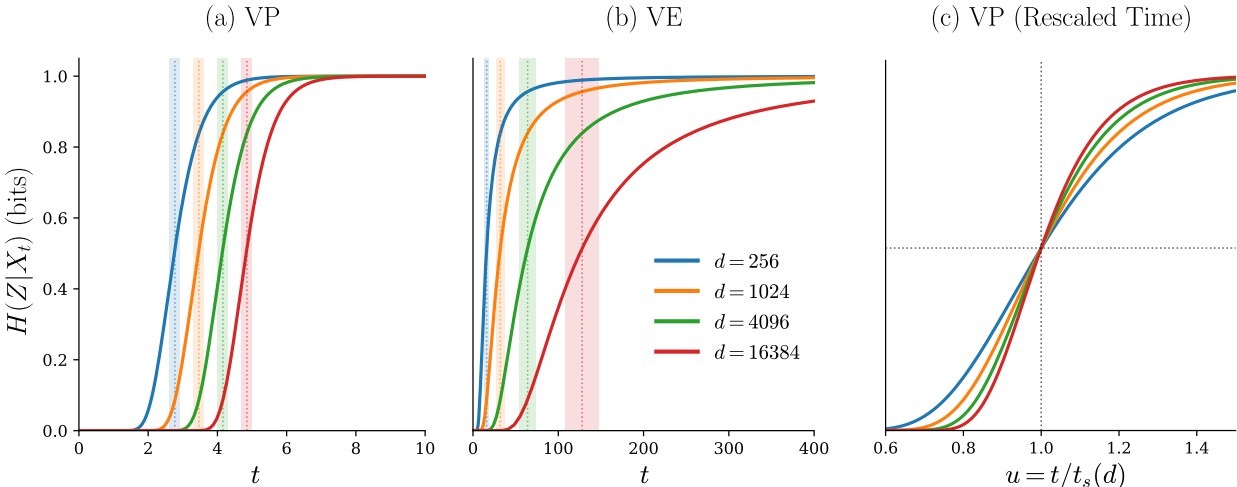

*Figure 2.* **Class-conditional entropy for an equiprobable two-component Gaussian mixture under VP and VE-EDM forward kernels.** Panels (a) and (b) show the entropy in physical time $t$ for the VP and VE-EDM kernels, respectively, across several dimensions $d$. Panel (c) shows the VP curves in the rescaled time variable $u = t/t_s$ with $t_s \sim \frac{1}{2}\log d$. Vertical dotted lines indicate the corresponding speciation scales. For VP, the transition sharpens into an $O(1)$ window around $u = 1$, whereas for VE-EDM the transition region broadens with $d$ on the physical time axis.

opposite happens: the posterior factor is no longer small, but the score gap has already collapsed. Since the posterior weight is always bounded by 1, the entire contribution again vanishes. Therefore, $\mathcal{E}_{i,k}(t)$ can only be appreciable in the intermediate regime $\lambda_{ik}(t) = O(1)$, where posterior competition has turned on while the score fields are still separated.

### 4.3. Speciation Time and VP/VE-EDM Scaling

We first consider the variance-preserving (VP) process, for which $\alpha_t = e^{-t}$ and $\sigma_t^2 = 1 - e^{-2t}$. In that case,

$$\lambda_{ik}(t) \asymp d\,\frac{e^{-2t}}{e^{-2t}\sigma_0^2 + 1 - e^{-2t}} \asymp de^{-2t}, \qquad (19)$$

so the nontrivial regime $\lambda_{ik}(t) = O(1)$ occurs at finite intervals around

$$t_s(d) = \frac{1}{2}\log d + O(1), \qquad (20)$$

which matches the speciation time predicted by Biroli et al. (2024). To see why this scale is special, rescale time as $u = t/t_s(d)$. Then $\lambda_{ik}(u) \asymp d^{1-u}$. For $u < 1$, we have $\lambda_{ik}(u) \to \infty$, while for $u > 1$, we instead have $\lambda_{ik}(u) \to 0$. It is therefore only in the critical regime $u = 1$ that posterior competition has become nontrivial while score separation has not yet disappeared. This shows that the entropy production concentrates in an $O(1)$ window around $t_s(d)$, or, equivalently, that it becomes increasingly localized around $u = 1$ as the dimension grows.

The same conclusion does not hold for VE-EDM-type kernels. In the EDM parameterization, $\alpha_t \equiv 1$ and $v(t) = \sigma_0^2 + t^2$, so that the control parameter yields the following speciation scale:

$$\lambda_{ik}(t) \asymp \frac{d}{\sigma_0^2 + t^2}, \qquad t_s(d) \asymp \sqrt{d}. \qquad (21)$$

However, after rescaling time as $u = t/t_s(d)$, $\lambda_{ik}$ remains of order $u^{-2}$. In other words, VE-EDM still has a characteristic scale, but unlike in the VP case, the transition does not sharpen around $u = 1$ as the dimension increases. The same balance between posterior competition and score separation still governs the dynamics, but the transition remains spread over a $O(1)$ range in rescaled time, which corresponds to a window of width $O(\sqrt{d})$ on the original time axis. Currently, the practical significance of this difference is unclear. Understanding whether it has implications for inference, training, or control remains an interesting direction for future work. The comparison is visible in Figure 2.

### 4.4. Extension to General Class Structure

The Gaussian calculation above is useful because its mechanism is explicit, but the same picture extends much more broadly. To make this precise, we now follow the general speciation framework of Achilli et al. (2026), which begins with a decomposition of the data distribution into semantically meaningful components,

$$p_0(a) = \sum_{r=1}^{N} w_r P_r(a), \qquad (22)$$

where the weights $w_r$ define the class prior and the distributions $P_r$ represent the individual classes. For the classes to be meaningful, we require that after adding a small but finite amount of noise, the originating class can still be identified with high probability by a Bayes classifier in the high-dimensional limit. This ensures that the labels correspond to genuinely distinguishable components of the data distribution, rather than to an arbitrary partition, and thus define a notion of class whose disappearance under diffusion can be meaningfully tracked. In addition, one assumes a self-averaging property for the class-conditioned log-likelihoods, so that the evidence carried by a typical sample can be separated into a deterministic extensive contribution and smaller sample-dependent fluctuations. Under these assumptions, if $X_t$ is generated from class $r$, then the likelihood under class $s$ can be written as

$$P_s(X_t; t) = \exp\big(d\, f_{rs}(t) + \delta f_{rs}(X_t, t)\big), \quad (23)$$

where $f_{rs}(t)$ is the deterministic intensive free-entropy and $\delta f_{rs}(X_t, t)$ collects the fluctuations around it.

In this setting, the pairwise log-odds take the form

$$\Lambda_{rs}(X_t, t) = \log \frac{w_r}{w_s} + d\big(f_{rr}(t) - f_{rs}(t)\big) \\ + \delta f_{rr}(X_t, t) - \delta f_{rs}(X_t, t). \quad (24)$$

This is analogous to the Gaussian decomposition in (14). The deterministic free-entropy gap plays the role of the average evidence in favor of class $r$, while the fluctuation term measures how strongly that evidence varies across samples. Achilli et al. (2026) show that speciation occurs when these two contributions become comparable in magnitude. This is precisely the time interval at which the entropy production peaks. For more details, consult Appendix A.4.

### 4.5. Partitioned Class-conditional Entropy

In realistic settings, the above discussion of speciation does not paint a full picture. From the pairwise decomposition in Eq. (8), we know that different class distinctions can contribute separately to the entropy-production profile. Consequently, semantic uncertainty can build up through several partially overlapping pairwise transitions rather than through a single isolated event. This is a limitation of the full class-conditional entropy: it aggregates all such branching events into a single observable and is therefore insensitive to which semantic distinction is responsible for a given increase in uncertainty. This is the behavior illustrated in panel (a) of Figure 1 and motivates the use of more targeted entropy probes.

A natural way to isolate specific branches is to impose a non-exhaustive binary partition on the set of classes. Let $Y_{\mathcal{S}}$ define this partition by a Bernoulli random variable given

two disjoint subsets of classes $(\mathcal{S}_0, \mathcal{S}_1)$, with $\mathcal{S}_0 \cup \mathcal{S}_1 = \mathcal{S} \subseteq \{1, \ldots, K\}$. Thus, we can write

$$Y_{\mathcal{S}} = \begin{cases} 0, & \text{if } Z \in \mathcal{S}_0, \\ 1, & \text{if } Z \in \mathcal{S}_1. \end{cases} \quad (25)$$

We can then define the corresponding *partitioned class-conditional entropy* by:

$$H(Y_{\mathcal{S}}|X_t) = -\int p_{t,\mathcal{S}}(x) \sum_{y \in \{0,1\}} \gamma_{y,\mathcal{S}}(x,t) \log \gamma_{y,\mathcal{S}}(x,t)\, \mathrm{d}x, \quad (26)$$

where $p_{t,\mathcal{S}}$ denotes the re-normalized mixture associated with the selected partition and $\gamma_{y,\mathcal{S}}(x,t)$ the corresponding binary posterior. This construction reduces the original $K$-ary inference problem to a binary one and turns the entropy into a probe of a specific semantic decision. As a result, partitioned class-conditional entropy can resolve branching events that would otherwise be blurred together inside the full class entropy and provide a principled tool for probing semantic distinctions at different levels of abstraction.

## 5. Numerical Experiments

In the following, we analyze the partitioned class-conditional entropy and its entropy production in trained models. The experiments were conducted using EDM2-XS (VE) and Stable Diffusion 1.5 (VP), trained on ImageNet-512 and LAION-5B, respectively.

### 5.1. Estimating the Entropy in Trained Models

In trained diffusion models, posteriors are not available in closed form. To estimate the partitioned class-conditional entropy $H(Y_{\mathcal{S}}|X_t)$ along the denoising trajectory, we employ an iterative procedure that utilizes the Markov structure of the forward diffusion process and the availability of conditional and unconditional models (Koulischer et al., 2025a). This allows us to propagate class posteriors backward along a denoising trajectory by iteratively updating log-posterior ratios using local likelihood increments between successive noise levels. Concretely, we estimate the posterior by accumulating the log-likelihood ratio between the conditional and reference denoisers. At each time step, the update is expressed in terms of squared reconstruction errors, which can be computed directly from the model's noise or noiseless data predictions (Algorithm 2 in Koulischer et al. (2025a)). The partitioned class-conditional entropy is then obtained by Monte Carlo averaging over the sampled trajectories. However, in the one-versus-complement setting, the induced binary prior is generally imbalanced, since $|\mathcal{S}_0| = 1$, while $|\mathcal{S}_1| = K - 1$ under equiprobable class priors. In T2I models, this imbalance is even stronger, as the class set is practically infinite. This leads to a suppression of the class-specific summand in Eq. 26 and the resulting Monte Carlo

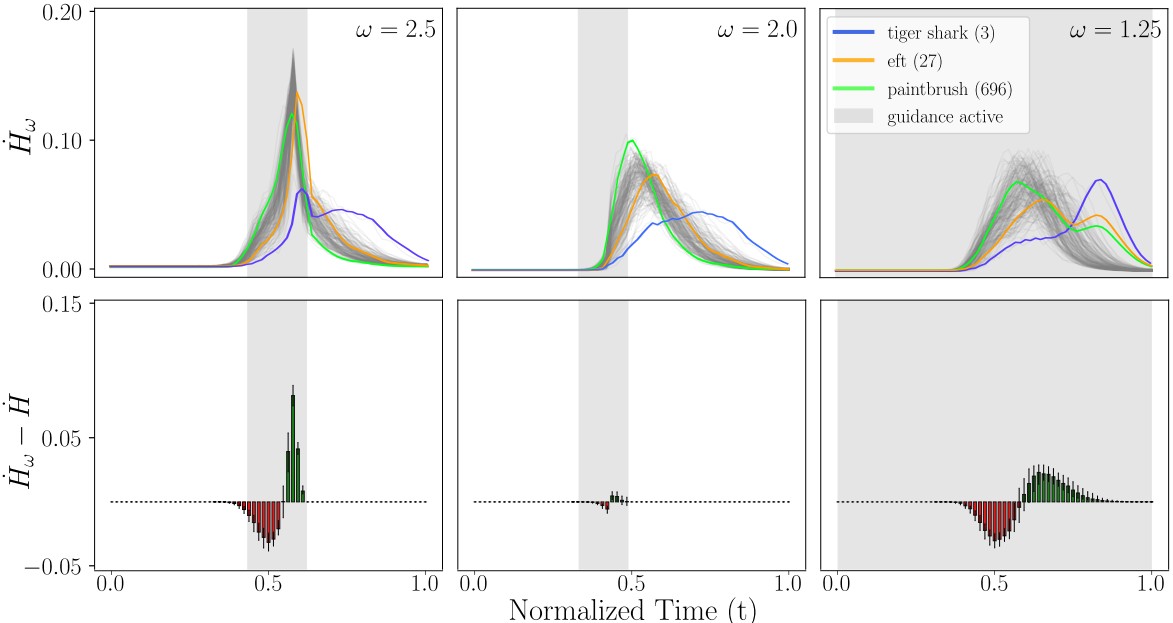

*Figure 3.* **Overview of information distortion caused by optimal guidance on ImageNet**. (Top) Partitioned class-conditional entropy production profiles when guidance with scale $\omega$ is applied within the gray interval. From left to right, intervals and guidance scales are optimized with respect to $\text{FD}_{\text{DINOv2}}$ (limited interval), FID (limited interval), and $\text{FD}_{\text{DINOv2}}$ with guidance applied throughout the full denoising trajectory. (Bottom) Difference between the guided entropy production $\frac{\partial}{\partial t} H_\omega(Y_{\mathcal{S}}|X_t)$ and the unguided baseline $\frac{\partial}{\partial t} H(Y_{\mathcal{S}}|X_t)$ (Figure 1), summarized by the median and the 25th and 75th percentiles across classes. Green bars indicate an increase in entropy production relative to the baseline, while red bars indicate a reduction.

estimate carries little signal. We combat this issue by setting the priors to 0.5 irrespective of the selected partition, leading to a Jenson-Shannon Divergence separability measure. It is important to note that this change does not have an effect on its quality as an operational measure of speciation, as it is still driven by pairwise competition between classes, thus acting according to the asymptotics presented in Section 4. In the one-versus-complement setting, we furthermore replace the complement model by the unconditional model, causing an estimation overlap that becomes negligible with increasing size of the class structure. We discuss these adaptions along with their effects on the estimator's robustness in Appendix B. The high-level description of our approximation procedure is given in Algorithm 1.

### 5.2. EDM2-XS

For ImageNet, we present two sets of results. First, the distribution of partitioned class-conditional entropy production profiles obtained from one-versus-complement partitions quantifying the class-specific speciation windows in the learned dynamics of EDM2-XS. Secondly, we study the effect of guidance on speciation as applied in practical settings. For this purpose, we first identify the empirically optimal guidance intervals according to FID (Heusel et al., 2017) and $\text{FD}_{\text{DINOv2}}$ (Stein et al., 2023), following (Kynkäänniemi et al., 2024) (Appendix C). We then estimate

---

**Algorithm 1** Estimating $H(Y_{\mathcal{S}}|X_t)$

1: **Input:** Conditional Denoiser $D_\theta$, $N$, $\tau$
2: Sample $x_\tau^{(0)}$, $x_\tau^{(1)}$ from $p_\tau$
3: $H_\tau \leftarrow 1$
4: **for** $y \in \{0, 1\}$ **do**
5:     Initialize $\gamma_y \leftarrow 0.5$
6:     **for** $t = \tau, \tau - 1, \ldots, 1$ **do**
7:         $x_{t-1}^{(y)} \leftarrow \text{sample}(y, D_\theta, x_t^{(y)}, t)$
8:         $\gamma_y \leftarrow \text{update\_posterior}(\gamma_y, D_\theta, x_t^{(y)}, x_{t-1}^{(y)}, t)$
9:         $h \leftarrow -(\gamma_y \log \gamma_y + (1 - \gamma_y) \log(1 - \gamma_y))$
10:         $I_{t-1}^{(y)} \leftarrow \frac{0.5}{N} \sum_{i=1}^{N} h^{(i)}$
11:     **end for**
12: **end for**
13: $H_{\tau-1:0} \leftarrow I_{\tau-1:0}^{(0)} + I_{\tau-1:0}^{(1)}$
14: **Output:** $H_{\tau:0}$

---

the partitioned class-conditional entropy production profiles under the guided trajectories and compare them to their non-guided counterparts. This yields a delta in information transfer that characterizes known pathologies of guidance, suggesting how guidance can accelerate speciation.

**Class-specific feature emergence** Consistent with the theoretical analysis in Section 4, we find that entropy production is confined to an interval in the intermediate stages

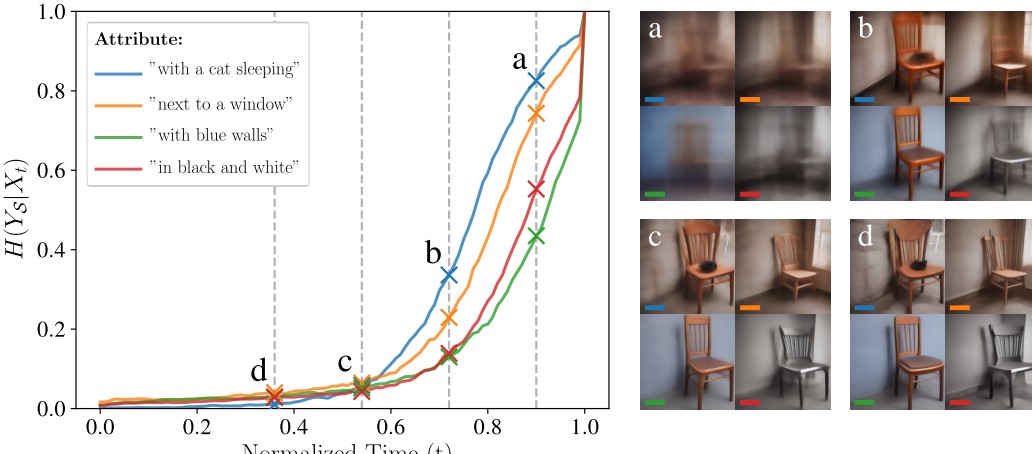

*Figure 4.* **Partitioned class-conditional entropy profiles for binary partitions of the form: "A wooden chair" vs. "A wooden chair + attribute"**. (Left) Profiles computed along the mixture distribution show that low-frequency changes (e.g., color) exhibit sharper entropy decay at higher noise levels. (Right) Samples with shared initial conditions confirm earlier generation of low-frequency details (blue walls) versus high-frequency elements (cat), with semantic commitment coinciding with entropy collapse.

of diffusion (Figure 1). A non-zero entropy production indicates regimes in which class posteriors fluctuate across samples, i.e., where semantic commitment is being resolved and class competition is still active. Empirically, most classes exhibit a similar unimodal profile centered around a common noise range, supporting previous claims of time-localized structure generation (Sclocchi et al., 2025; Li & Chen, 2024). At the same time, there are systematic class-dependent deviations in the location and shape of this window. For example, *tiger shark* exhibits an entropy production profile that is shifted toward higher noise levels and sustained over a broader interval, whereas *paintbrush* peaks at substantially lower noise. This indicates that different classes resolve their defining semantics at different effective noise scales. In particular, peaks at high noise levels are consistent with semantics captured by coarse, low-order statistics of the conditional distribution, while peaks at lower noise levels suggest reliance on higher-frequency, fine-grained structure which is supported by the denoised predictions shown in the lower half of Figure 1 and additional examples provided in Appendix C.

**Information distortion under guidance** Figure 3 shows that guidance systematically redistributes partitioned class-conditional entropy production $\frac{\partial}{\partial t} H_w(Y_{\mathcal{S}}|X_t)$ along the denoising trajectory relative to their unguided baselines from Figure 1. Guidance restricted to the FD$_{\text{DINOv2}}$-optimal interval shifts entropy production toward higher noise levels and suppresses it at lower ones, thereby accelerating overall speciation for most classes. When guidance is applied within the FID-optimal interval (center column), we do not observe such a strong effect because the entropy production has already mostly collapsed. Thus, guidance in this interval mainly affects within-class speciation or feature refine-

ment, so the collapse onto individual data points. Applied throughout the entire trajectory (right column), guidance produces a pronounced shift of entropy production toward high-noise stages. The corresponding information shift remains relatively uniform, consistent with a gradual denoising of coarse, low-frequency structure rather than a sharp semantic transition. This redistribution is most pronounced for classes whose unguided entropy production overlaps with the guided interval, such as *tiger shark*, and largely absent for classes whose entropy production is mainly limited to low noise levels. In order to make these effects visually more accessible to the reader, we provide generated images along with predicted noiseless trajectories (under guidance) for the three highlighted classes in Appendix C.

### 5.3. Stable Diffusion 1.5

In ImageNet, the emergence of semantic structure can only be probed at the abstraction level defined by the latent class variable. Therefore, the interpretation of the corresponding entropy profiles remains agnostic to finer semantic attributes within a class and consequently the probing of speciation. Text-conditioned diffusion models provide a substantially more informative semantic variable, enabling the estimation of Eq. (26) for partitions that target specific and localized semantic content. In these settings, the entropy can be interpreted as a measure of the overlap between the marginal distributions of two prompts, with high and low overlap corresponding to high and low entropy, respectively. This overlap diminishes as semantic differences between the prompts are resolved during the reverse process, thereby isolating the temporal window in which the variant-specific feature emerges. To measure the emergence of specific attributes, we consider base prompts together with minimally modi-

fied variants designed to introduce semantic differences at different levels of abstraction.

Figure 4 shows representative entropy profiles for four prompt variants that target semantic attributes at different spatial and spectral scales. Low-frequency decisions, such as global color changes (e.g., *"with blue walls"* or *"in black and white"*), exhibit an early and rapid entropy decay, indicating semantic commitment at high noise levels. In contrast, localized, high-frequency attributes, such as the presence of a *cat* or a *window*, show a slower entropy collapse, consistent with semantic resolution at later stages of denoising. Figure 4 visualizes this correspondence by showing noiseless predictions at selected timesteps for each variant. In snapshot (a), decisions regarding the blue walls and the black-and-white aesthetics appear resolved, while substantial ambiguity remains over the window and the cat. In (b), both low-frequency components are fully determined and the window structure begins to emerge. Only in (c) have all semantic decisions been made, as reflected by the collapse of all entropy profiles. Additional examples for the same base prompt and other prompt configurations are provided and discussed in Appendix C.

## 6. Limitations and Future Work

Our analysis relies on an online posterior update mechanism that compares conditional and reference noise predictions using local likelihood increments. Its accuracy depends on the calibration of these predictions across noise levels. Systematic prediction errors can bias posterior updates and distort entropy profiles, particularly when models are probed out of distribution. In the ImageNet setting, where the class space is well defined, using the unconditional model as a proxy for the class complement provides a stable estimator in practice. However, this approximation can introduce bias when the proxy deviates from the true complement distribution.

For Stable Diffusion 1.5, the present experiments are further limited by the model's ability to generate fine-grained detail. Prompt modifications can induce larger distributional shifts than intended, complicating the isolation of individual semantic attributes. Extending the analysis to stronger and better-aligned models, such as Stable Diffusion XL or Stable Diffusion 3, is therefore a natural next step (Rombach et al., 2022; Esser et al., 2024). More broadly, future work should explore the use of external models trained to detect specific features as part of the estimation pipeline. Finally, a more formal characterization of how guidance controls the magnitude and timing of entropy redistribution along the sampling trajectory remains an open problem.

## 7. Conclusion

We introduced class-conditional entropy and its temporal derivative as practical tools for analyzing semantic emergence in diffusion models. In high-dimensional Gaussian mixtures, we showed that entropy production concentrates on the same logarithmic time scale as the speciation instability predicted by statistical physics, linking symmetry breaking to an information-theoretic measure of uncertainty. Experiments on EDM2-XS and Stable Diffusion 1.5 demonstrate that partitioned entropy reliably localizes the noise regimes in which semantic distinctions are resolved and provides a quantitative view of how guidance redistributes semantic information over time, consistent with recent findings on its time-dependent effectiveness (Kynkäänniemi et al., 2024). Together, these results connect information-theoretic and dynamical perspectives on diffusion and offer a principled diagnostic for time-localized analysis of guided generative models.

## Acknowledgments

This research was funded by the imec.prospect project ADAPT, a research project bringing together academic researchers and industry partners and financed by imec, the Research Foundation Flanders (FWO-Vlaanderen) under grant G0C2723N, and the Flemish Government under the "Onderzoeksprogramma Artificiele Intelligentie (AI) Vlaanderen" programme.

## Impact statement

This paper presents work whose goal is to advance the field of machine learning. There are many potential societal consequences of our work, none of which we feel must be specifically highlighted here.

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

## A. Asymptotic Analysis of the Class-conditional Entropy for a Gaussian Mixture

In this section, we show that the class-conditional entropy production provides a signature of speciation. Starting from a general identity that expresses entropy production in terms of posterior class competition and pairwise score separation, we identify the mechanism behind the transition. We then make this mechanism explicit in isotropic Gaussian mixtures, where both the posterior log-odds and the score gaps are tractable, allowing a comparison between the VP and VE/EDM forward SDEs. Finally, we return to the VP setting and adopt the same perspective to connect our entropy-production picture to the general theory of speciation for arbitrary class structure.

### A.1. Partitioned- and Conditional Entropy Production

Here, we derive the result given in Eq. (8) and extend it to Eq. (26) and its entropy production form. First, we note that we can use a standard information theoretic identity to write the class-conditional entropy as:

$$H(Z|X_t) = H(Z) - I(X_t; Z) = H(Z) - h(X_t) + h(X_t|Z). \tag{27}$$

Taking the derivative with respect to time $t$ eliminates $H(Z)$. Therefore:

$$\frac{\partial}{\partial t} H(Z|X_t) = \frac{\partial}{\partial t} h(X_t|Z) - \frac{\partial}{\partial t} h(X_t). \tag{28}$$

Furthermore, $h(X_t)$ and $h(X_t|Z)$ are equivalent in form. Thus, it suffices to derive the result for either one and extend to the other. Starting from the definition of $h(X_t|Z)$ and defining $w_k := p(Z = k)$:

$$
\begin{aligned}
h(X_t|Z) &= -\sum_{k=1}^{K} w_k \int p_t(x|Z = k) \log p_t(x|Z = k) \, \mathrm{d}x \\
&= \sum_{k=1}^{K} w_k \, h(X_t|Z = k).
\end{aligned}
\tag{29}
$$

Taking the derivative with respect to $t$ then gives:

$$\frac{\partial}{\partial t} h(X_t|Z = k) = -\int \frac{\partial}{\partial t} p_t(x|Z = k) \log p_t(x|Z = k) \, \mathrm{d}x. \tag{30}$$

Assuming that the process is described by Eq. (1), we can further simplify using the Fokker-Planck equation (2):

$$
\begin{aligned}
\frac{\partial}{\partial t} h(X_t|Z = k) &= -\int \frac{\partial}{\partial t} p_t(x|Z = k) \log p_t(x|Z = k) \, \mathrm{d}x \\
&= -\int \left[ -\nabla \cdot \left( \left( f(x, t) - \frac{g_t^2}{2} \nabla \log p_t(x|Z = k) \right) p_t(x|Z = k) \right) \right] \log p_t(x|Z = k) \, \mathrm{d}x \\
&= -\int \left\langle f(x, t) - \frac{g_t^2}{2} \nabla \log p_t(x|Z = k), \nabla \log p_t(x|Z = k) \right\rangle p_t(x|Z = k) \, \mathrm{d}x \\
&= -\int \langle f(x, t), \nabla \log p_t(x|Z = k) \rangle \, p_t(x|Z = k) \, \mathrm{d}x + \frac{g_t^2}{2} \int \|\nabla \log p_t(x|Z = k)\|_2^2 \, p_t(x|Z = k) \, \mathrm{d}x \\
&= \int (\nabla \cdot f(x, t)) \, p_t(x|Z = k) \, \mathrm{d}x + \frac{g_t^2}{2} \int \|\nabla \log p_t(x|Z = k)\|_2^2 \, p_t(x|Z = k) \, \mathrm{d}x.
\end{aligned}
\tag{31}
$$

Similarly, for the marginal entropy, we get:

$$\frac{\partial}{\partial t} h(X_t) = \int (\nabla \cdot f(x, t)) \, p_t(x) \, \mathrm{d}x + \frac{g_t^2}{2} \int \|\nabla \log p_t(x)\|_2^2 \, p_t(x) \, \mathrm{d}x. \tag{32}$$

Because divergence terms cancel, $p_t(x) = \sum_{k=1}^{K} p_t(x|Z = k)p(Z = k)$, and the squared norms can be combined under the joint, we obtain:

$$\frac{\partial}{\partial t} H(Z|X_t) = \frac{g_t^2}{2} \sum_{k=1}^{K} w_k \int p_t(x|Z = k) \left( ||\nabla \log p_t(x|Z = k)||_2^2 - ||\nabla \log p_t(x)||_2^2 \right) dx$$

$$= \frac{g_t^2}{2} \sum_{k=1}^{K} w_k \int p_t(x|Z = k) ||\nabla \log p_t(x|Z = k) - \nabla \log p_t(x)||_2^2 \, dx$$

(33)

Using the score identity:

$$s_{\text{mix}}(x, t) := \nabla \log p_t(x) = \sum_{k=1}^{K} \gamma_k(x, t) \, s_k(x, t),$$

$$\gamma_k(x, t) := \frac{w_k \, p_t(x|Z = k)}{p_t(x)}, \qquad s_k(x, t) := \nabla \log p_t(x|Z = k),$$

(34)

we can furthermore write the squared norm as a sum over pairwise score gaps. Starting from the previous result and writing it in a slightly altered form:

$$\frac{\partial}{\partial t} H(Z|X_t) = \frac{g_t^2}{2} \int p_t(x) \sum_{r=1}^{K} \gamma_r(x, t) ||s_r(x, t) - s_{\text{mix}}||_2^2 \, dx,$$

$$= \frac{g_t^2}{2} \int p_t(x) \left( \sum_{r=1}^{K} \gamma_r(x, t) ||s_r(x, t)||_2^2 - 2 \sum_{r=1}^{K} \gamma_r(x, t) \langle s_r(x, t), s_{\text{mix}}(x, t) \rangle + \sum_{r=1}^{K} \gamma_r(x, t) ||s_{\text{mix}}(x, t)||_2^2 \right) dx$$

$$= \frac{g_t^2}{2} \int p_t(x) \left( \sum_{r=1}^{K} \gamma_r(x, t) ||s_r(x, t)||_2^2 - \sum_{r=1}^{K} \gamma_r(x, t) \langle s_r(x, t), s_{\text{mix}}(x, t) \rangle \right) dx$$

$$= \frac{g_t^2}{2} \int p_t(x) \left( \sum_{s=1}^{K} \gamma_s(x, t) \sum_{r=1}^{K} \gamma_r(x, t) ||s_r(x, t)||_2^2 - \left\langle \sum_{r=1}^{K} \gamma_r(x, t) s_r(x, t), \sum_{s=1}^{K} \gamma_s(x, t) s_s(x, t) \right\rangle \right) dx$$

$$= \frac{g_t^2}{2} \int p_t(x) \left( \sum_{r,s=1}^{K} \gamma_s(x, t) \gamma_r(x, t) ||s_r(x, t)||_2^2 - \sum_{r,s=1}^{K} \gamma_s(x, t) \gamma_r(x, t) \langle s_r(x, t), s_s(x, t) \rangle \right) dx$$

$$= \frac{g_t^2}{2} \int p_t(x) \frac{1}{2} \sum_{r,s=1}^{K} \gamma_s(x, t) \gamma_r(x, t) ||s_r(x, t) - s_s(x, t)||_2^2 \, dx$$

$$= \frac{g_t^2}{4} \sum_{r,s=1}^{K} w_r \int p_t(x|Z = r) \, \gamma_s(x, t) ||s_r(x, t) - s_s(x, t)||_2^2 \, dx.$$

(35)

which is the general result defined in Eq. (8) with:

$$\mathcal{E}_{r,s}(t) := \int p_t(x|Z = r) \gamma_s(x, t) \, ||s_r(x, t) - s_s(x, t)||_2^2 \, dx.$$

(36)

Following the same logic, we can write the partitioned class-conditional entropy production for a binary partition $\mathcal{S} = \mathcal{S}_0 \cup \mathcal{S}_1$, with $\mathcal{S}_0 \cap \mathcal{S}_1 = \varnothing$, as:

$$\frac{\partial}{\partial t} H(Y_{\mathcal{S}}|X_t) = \frac{g_t^2}{2} \int p_{t,\mathcal{S}}^{\pi}(x) \sum_{i \in \{0,1\}} \gamma_{i,\mathcal{S}}^{\pi}(x, t) ||s_{i,\mathcal{S}}(x, t) - s_{\text{mix},\mathcal{S}}^{\pi}(x, t)||_2^2 \, dx$$

$$= \frac{g_t^2}{2} \sum_{i \in \{0,1\}} \pi_i \int q_{i,\mathcal{S}}(x, t) \left( \gamma_{1-i,\mathcal{S}}^{\pi}(x, t) \right)^2 ||s_{0,\mathcal{S}}(x, t) - s_{1,\mathcal{S}}(x, t)||_2^2 \, dx,$$

(37)

where we have added the superscript $\pi$ to facilitate the discussion of the implementation details in Section B:

$$q_{0,\mathcal{S}}(x,t) := p_t(x|Y_{\mathcal{S}}=0) := \sum_{k \in \mathcal{S}_0} p_t(x|Z=k) \frac{w_k}{\sum_{j \in \mathcal{S}_0} w_j}, \qquad q_{1,\mathcal{S}}(x,t) := p_t(x|Y_{\mathcal{S}}=1) := \sum_{k \in \mathcal{S}_1} p_t(x|Z=k) \frac{w_k}{\sum_{j \in \mathcal{S}_1} w_j},$$

$$p_{t,\mathcal{S}}^{\pi}(x) := \pi_0 q_{0,\mathcal{S}}(x,t) + \pi_1 q_{1,\mathcal{S}}(x,t), \qquad \gamma_{i,\mathcal{S}}^{\pi}(x,t) := \frac{\pi_i q_{i,\mathcal{S}}(x,t)}{p_{t,\mathcal{S}}^{\pi}(x)}, \qquad i \in \{0,1\},$$

$$s_{i,\mathcal{S}}(x,t) := \nabla_x \log q_{i,\mathcal{S}}(x,t), \qquad s_{\text{mix},\mathcal{S}}^{\pi}(x,t) := \nabla_x \log p_{t,\mathcal{S}}^{\pi}(x) = \sum_{i \in \{0,1\}} \gamma_{i,\mathcal{S}}^{\pi}(x,t) s_{i,\mathcal{S}}(x,t),$$

$$\pi_0 := \frac{\sum_{k \in \mathcal{S}_0} w_k}{\sum_{j \in \mathcal{S}} w_j}, \qquad \pi_1 := 1 - \pi_0.$$

$$(38)$$

### A.2. General Entropy-production Identity

Let $Z \in \{1, \dots, K\}$ denote the class label. Starting from the definition of the class-conditional entropy production derived in the previous section:

$$\frac{\partial}{\partial t} H(Z|X_t) = \frac{g_t^2}{4} \sum_{r,s=1}^{K} w_r \int p_t(x|Z=r) \gamma_s(x,t) \|s_r(x,t) - s_s(x,t)\|_2^2 \, dx, \qquad (39)$$

we define for each ordered pair $(r,s)$, the posterior log-odds:

$$\Lambda_{rs}(x,t) := \log \frac{\gamma_r(x,t)}{\gamma_s(x,t)}. \qquad (40)$$

Then, for a fixed source class $Z = r$, we get:

$$\gamma_r(x,t) = \frac{1}{1 + \sum_{u \neq r} e^{-\Lambda_{ru}(x,t)}}, \qquad \gamma_s(x,t) = \frac{e^{-\Lambda_{rs}(x,t)}}{1 + \sum_{u \neq r} e^{-\Lambda_{ru}(x,t)}} \quad (s \neq r). \qquad (41)$$

Substituting this into the entropy production gives:

$$\frac{\partial}{\partial t} H(Z|X_t) = \frac{g_t^2}{4} \sum_{r=1}^{K} w_r \int p_t(x|Z=r) \sum_{s \neq r} \frac{e^{-\Lambda_{rs}(x,t)}}{1 + \sum_{u \neq r} e^{-\Lambda_{ru}(x,t)}} \|s_r(x,t) - s_s(x,t)\|_2^2 \, dx. \qquad (42)$$

This result makes the entropy-production mechanism transparent. For samples originating from class $r$, the contribution is large only in regions where competing classes $s \neq r$ retain non-negligible posterior mass, as quantified by the softmax factors while their local score fields remain well separated from that of class $r$, as quantified by $\|s_r(x,t) - s_s(x,t)\|_2^2$. Thus, entropy production is driven by the joint presence of posterior ambiguity and geometric disagreement in score space. The effect is additive across competitors. A large peak may arise either from a single dominant rival class or from the collective contribution of an entire block of classes whose posterior masses remain appreciable and whose score fields are still distinct from that of class $r$.

### A.3. Entropy Production in Isotropic Gaussian Mixtures

We now specialize to the solvable case of an equiprobable isotropic Gaussian mixture. Let the Gaussian forward kernel along with the equiprobable mixture be given by:

$$p(x_t|x_0) = \mathcal{N}(x_t| \alpha_t x_0, \sigma_t^2 I_d), \qquad p_0(x) = \frac{1}{K} \sum_{k=1}^{K} \mathcal{N}(x|\mu_k, \sigma_0^2 I_d). \qquad (43)$$

We assume that the means satisfy the high-dimensional scaling:

$$\frac{\|\mu_k\|_2^2}{d} = q_k > 0, \qquad \frac{\|\mu_k - \mu_i\|_2^2}{d} = \delta_{ik}^2 > 0, \qquad i \neq k, \qquad (44)$$

with $\sigma_0 = O(1)$ and a fixed number of classes $K$. Under the given forward kernel, each class remains Gaussian and, hence, the marginal at time $t$ is defined via the mixture:

$$p_t(x) = \frac{1}{K} \sum_{k=1}^{K} \mathcal{N}\big(x|m_k(t), v(t)\mathrm{I}_d\big), \qquad m_k(t) = \alpha_t \mu_k, \qquad v(t) = \alpha_t^2 \sigma_0^2 + \sigma_t^2. \tag{45}$$

The class-conditional and mixture score fields alongside the responsibilities are accordingly given by:

$$s_k(x,t) = \nabla_x \log p_t(x|Z=k) = -\frac{1}{v(t)}\big(x - m_k(t)\big)$$

$$s_{\mathrm{mix}}(x,t) = \sum_{k=1}^{K} \gamma_k(x,t)\, s_k(x,t) = -\frac{1}{v(t)}\left(x - \sum_{k=1}^{K} \gamma_k(x,t)m_k(t)\right) \tag{46}$$

$$\gamma_k(x,t) = \frac{\exp\big(-\frac{1}{2v(t)}\|x - m_k(t)\|_2^2\big)}{\sum_{\ell=1}^{K} \exp\big(-\frac{1}{2v(t)}\|x - m_\ell(t)\|_2^2\big)}.$$

For every pair $(r,s)$, the score difference and according squared norm are particularly simple:

$$s_r(x,t) - s_s(x,t) = \frac{m_r(t) - m_s(t)}{v(t)}, \qquad \|s_r(x,t) - s_s(x,t)\|_2^2 = \frac{\|m_r(t) - m_s(t)\|_2^2}{v(t)^2}, \tag{47}$$

where the squared norm of the score difference is deterministic in $x$. So, in the isotropic Gaussian case, the randomness in the entropy production enters only through the posterior factors which follow from the pairwise posterior log-odds given by:

$$\Lambda_{rs}(x,t) = \frac{\|x - m_s(t)\|_2^2 - \|x - m_r(t)\|_2^2}{2v(t)}$$

$$= \frac{\|m_s(t)\|_2^2 - \|m_r(t)\|_2^2 + 2x^\top\big(m_r(t) - m_s(t)\big)}{2v(t)}. \tag{48}$$

Conditioning on a source class $Z = r$, that is, $X_t = m_r(t) + \sqrt{v(t)}\epsilon$, with $\epsilon \sim \mathcal{N}(0, \mathrm{I}_d)$, we can further write them as:

$$\Lambda_{rs} = \frac{\|m_s(t)\|_2^2 - \|m_r(t)\|_2^2}{2v(t)} + \frac{2\big(m_r(t) + \sqrt{v(t)}\epsilon\big)^\top\big(m_r(t) - m_s(t)\big)}{2v(t)}$$

$$= \frac{\|m_s(t)\|_2^2 - \|m_r(t)\|_2^2 + 2\|m_r(t)\|_2^2 - 2m_r(t)^\top m_s(t)}{2v(t)} + \frac{\epsilon^\top\big(m_r(t) - m_s(t)\big)}{\sqrt{v(t)}} \tag{49}$$

$$= \frac{\|m_s(t) - m_r(t)\|_2^2}{2v(t)} + \frac{\big(m_r(t) - m_s(t)\big)^\top\epsilon}{\sqrt{v(t)}}.$$

Consequently, under the conditional sampling law, $\Lambda_{rs}$ is Gaussian with mean and variance given by:

$$\mathbb{E}[\Lambda_{rs}(X_t,t) \mid Z=r] = \frac{\|m_s(t) - m_r(t)\|_2^2}{2v(t)}, \qquad \mathrm{Var}(\Lambda_{rs}(X_t,t) \mid Z=r) = \frac{\|m_s(t) - m_r(t)\|_2^2}{v(t)}. \tag{50}$$

It is therefore natural to introduce the pairwise control parameter:

$$\lambda_{rs}(t) := \frac{\|m_r(t) - m_s(t)\|_2^2}{v(t)} = \frac{\alpha_t^2\|\mu_r - \mu_s\|_2^2}{\alpha_t^2\sigma_0^2 + \sigma_t^2}, \qquad \Lambda_{rs}(X_t,t) = \frac{\lambda_{rs}(t)}{2} + \sqrt{\lambda_{rs}(t)}\,\xi_{rs},$$

$$\xi_{rs} := \frac{\big(m_r(t) - m_s(t)\big)^\top\epsilon}{\|m_r(t) - m_s(t)\|_2}, \qquad \epsilon \sim \mathcal{N}(0, \mathrm{I}_d), \qquad \xi_{rs} \sim \mathcal{N}(0,1). \tag{51}$$

The deterministic part of the log-odds is $\lambda_{rs}(t)/2$, while the fluctuation scale is $\sqrt{\lambda_{rs}(t)}$. The Gaussian speciation window is precisely the regime in which $\lambda_{rs}(t) = O(1)$ for a nontrivial block of competing classes. Using the previous results, the

entropy production becomes:

$$\frac{\partial}{\partial t} H(Z|X_t) = \frac{g_t^2}{4K} \sum_{r=1}^{K} \int p_t(x|Z=r) \sum_{s \neq r} \frac{e^{-\Lambda_{rs}(x,t)}}{1 + \sum_{u \neq r} e^{-\Lambda_{ru}(x,t)}} \|s_r(x,t) - s_s(x,t)\|_2^2 \, dx$$

$$= \frac{g_t^2}{4K} \sum_{r=1}^{K} \int p_t(x|Z=r) \sum_{s \neq r} \frac{e^{-\Lambda_{rs}(x,t)}}{1 + \sum_{u \neq r} e^{-\Lambda_{ru}(x,t)}} \frac{\|m_r(t) - m_s(t)\|_2^2}{v(t)^2} \, dx \qquad (52)$$

$$= \frac{g_t^2}{4K \, v(t)} \sum_{r=1}^{K} \int p_t(x|Z=r) \sum_{s \neq r} \frac{e^{-\Lambda_{rs}(x,t)}}{1 + \sum_{u \neq r} e^{-\Lambda_{ru}(x,t)}} \lambda_{rs}(t) \, dx$$

Again, the entropy-production mechanism around speciation becomes apparent. Before speciation, all relevant $\Lambda_{rs}$ are typically large and positive under source class $r$, so the total competitor mass is negligible. After speciation, the posterior becomes nontrivial, but the score gaps collapse. At the transition, a whole block of classes can simultaneously satisfy $\lambda_{rs}(t) = O(1)$, leading to an $O(1)$ competitor mass while the score gaps are still macroscopic. It is important to note that both the contribution of geometry and the competition between classes is controlled by pairwise $\lambda_{rs}$.

### Speciation in Variance-preserving OU-processes

For the variance-preserving OU process, we have $\alpha_t = e^{-t}$ and $\sigma_t^2 = 1 - e^{-2t}$. The control parameter therefore becomes:

$$v(t) := \alpha_t^2 \sigma_0^2 + \sigma_t^2 = e^{-2t}\sigma_0^2 + (1 - e^{-2t}), \qquad m_k(t) := \alpha_t \mu_k = e^{-t}\mu_k,$$

$$\lambda_{rs}(t) = \frac{\|m_r(t) - m_s(t)\|_2^2}{v(t)} = \frac{\|e^{-t}(\mu_r - \mu_s)\|_2^2}{e^{-2t}\sigma_0^2 + (1 - e^{-2t})} \qquad (53)$$

Under the scaling assumption $\|\mu_r - \mu_s\|_2^2 \asymp d$, the control parameter is bounded by:

$$\frac{\beta_1 d e^{-2t}}{e^{-2t}\sigma_0^2 + 1 - e^{-2t}} \leq \lambda_{rs}(t) \leq \frac{\beta_2 d e^{-2t}}{e^{-2t}\sigma_0^2 + 1 - e^{-2t}}, \qquad \beta_1, \beta_2 > 0. \qquad (54)$$

And because its denominator does not scale with $d$, we can further restrict it:

$$\min(\sigma_0^2, 1) \leq e^{-2t}\sigma_0^2 + 1 - e^{-2t} \leq \max(\sigma_0^2, 1), \qquad \frac{\beta_1 d e^{-2t}}{\max(\sigma_0^2, 1)} \leq \lambda_{rs}(t) \leq \frac{\beta_2 d e^{-2t}}{\min(\sigma_0^2, 1)}. \qquad (55)$$

The speciation timescale is obtained by requiring $\lambda_{rs}(t_s(d)) \asymp 1$, which marks the regime where the pairwise log-odds are non-trivial. This leads to the speciation time from Biroli et al. (2024):

$$\lambda_{rs}(t) \asymp d e^{-2t}, \qquad t_s(d) = \frac{1}{2} \log d + O(1) \qquad (56)$$

Next, we will show that the entropy production concentrates on the given time scale and becomes negligible before and after. For this reason, we introduce the rescaled time into the control parameter:

$$u := \frac{t}{t_s(d)}, \qquad m_k(u) = d^{-u/2}\mu_k, \qquad v(u) = 1 + (\sigma_0^2 - 1)d^{-u},$$

$$\Rightarrow \lambda_{rs}(u) = \frac{\|\mu_r - \mu_s\|_2^2}{d^u + \sigma_0^2 - 1} \qquad (57)$$

Equivalently, under source class $Z = r$, and again using the aforementioned mean scaling, we get that the growth of mean and variance of the pairwise log-odds behaves according to:

$$\mathbb{E}[\Lambda_{rs}(X_t, u) \mid Z = r] = \frac{\|\mu_r - \mu_s\|_2^2}{2(d^u + \sigma_0^2 - 1)} \asymp \frac{d}{d^u} = d^{1-u}, \qquad \text{Var}[\Lambda_{rs}(X_t, u) \mid Z = r] = \frac{\|\mu_r - \mu_s\|_2^2}{d^u + \sigma_0^2 - 1} \asymp d^{1-u}. \qquad (58)$$

We now distinguish three regimes: $u < 1$, $u = 1$, and $u > 1$. We will therefore study the control parameter and its effect on the pairwise log-odds as $d \to \infty$. Again, using the scaling assumption for the separation of means, we get:

$$\lambda_{rs}(u) = \frac{\|\mu_r - \mu_s\|_2^2}{d^u + \sigma_0^2 - 1} \asymp d^{1-u}, \qquad \Lambda_{rs}(X_t, u) \mid (Z = r) \sim \mathcal{N}\left(\frac{\lambda_{rs}(u)}{2}, \lambda_{rs}(u)\right). \tag{59}$$

For $u < 1$, $\lambda_{rs}(u)$ becomes arbitrarily large. This behavior translates to $\Lambda_{rs}(X_t, u)$ through both its mean and its fluctuations. After normalization by $\lambda_{rs}(u)$, the mean equals $1/2$ while the variance tends to $0$:

$$\mathbb{E}\left[\frac{\Lambda_{rs}(X_t, u)}{\lambda_{rs}(u)} \,\middle|\, Z = r\right] = \frac{1}{2}, \qquad \mathrm{Var}\left[\frac{\Lambda_{rs}(X_t, u)}{\lambda_{rs}(u)} \,\middle|\, Z = r\right] = \frac{1}{\lambda_{rs}} \to 0. \tag{60}$$

This implies that with high probability $\Lambda_{rs}$ is positive. We can quantify this behavior even further by using the exponential tail bound given by:

$$\mathbb{P}\big(L \le -a\big) \le e^{-ca^2}, \qquad L \sim \mathcal{N}(0, 1), \qquad c > 0,$$

$$\mathbb{P}\left(\Lambda_{rs}(X_t, u) \le \frac{\lambda_{rs}(u)}{4} \,\middle|\, Z = r\right) = \mathbb{P}\left(\xi_{rs} \le -\frac{\sqrt{\lambda_{rs}(u)}}{4} \,\middle|\, Z = r\right) \le e^{-c'\lambda_{rs}(u)}, \qquad c' = c/16 > 0, \tag{61}$$

stating that with high probability the log-odds remain at least a positive fraction of the mean, here $1/2$. Therefore, the posterior mass of a fixed competing class $r \ne s$ is exponentially small, i.e., $\gamma_s(X_t, u) \le e^{-\lambda_{rs}(u)/4}$. We can generalize this statement to the totality of competing posterior mass by using the union bound, which gives:

$$\mathbb{P}\left(\exists\, s \ne r : \Lambda_{rs}(X_t, u) \le \frac{1}{2}\frac{\lambda_{rs}(u)}{2} \,\middle|\, Z = r\right) \le \sum_{s \ne r} \mathbb{P}\left(\Lambda_{rs}(X_t, u) \le \frac{1}{2}\frac{\lambda_{rs}(u)}{2} \,\middle|\, Z = r\right) \le \sum_{s \ne r} e^{-c'\lambda_{rs}(u)}. \tag{62}$$

Using $\Lambda_{rs}(X_t, u) \ge \frac{\lambda_{rs}(u)}{4}$, we thus obtain the bound on the total competing posterior mass given by:

$$\sum_{s \ne r} \gamma_s(X_t, u) = \sum_{s \ne r} \frac{e^{-\Lambda_{rs}(X_t, u)}}{1 + \sum_{j \ne r} e^{-\Lambda_{rj}(X_t, u)}} \le \sum_{s \ne r} e^{-\lambda_{rs}(u)/4}. \tag{63}$$

At the same time, we observe that the score gap is polynomial in $d$ :

$$\|s_r(X_t, u) - s_s(X_t, u)\|_2^2 = \frac{\lambda_{rs}(u)}{v(u)} \asymp d^{1-u}. \tag{64}$$

Given the definition of the class-conditional entropy production:

$$\frac{\partial}{\partial t} H(Z|X_t) = \frac{g_t^2}{4K\,v(u)} \sum_{r=1}^{K} \int p_t(x|Z = r) \sum_{s \ne r} \gamma_s(x, u)\lambda_{rs}(u)\,\mathrm{d}x, \tag{65}$$

we can see that for the high probability event $\Lambda_{rs}(X_t, u) \ge \frac{\lambda_{rs}(u)}{4}$, each individual bound in the inner summand is controlled by a polynomial term multiplied by an exponentially small term, and therefore in the limit $d \to \infty$ the entropy decays to zero:

$$\sum_{s \ne r} e^{-\lambda_{rs}(u)/4}\lambda_{rs}(u) = O(d^{1-u}e^{-\kappa_1 d^{1-u}/4}), \qquad \frac{\partial}{\partial t} H(Z|X_t) \to 0, \tag{66}$$

which holds for every $u < 1$ and since $K$ is fixed.

For $u > 1$, the control parameter satisfies

$$\lambda_{rs}(u) \asymp d^{1-u} \to 0. \tag{67}$$

In this regime the posterior competition may remain nontrivial, but this no longer produces entropy because the class-conditional score fields have already collapsed. Indeed, for the isotropic Gaussian mixture,

$$\|s_r(X_t, u) - s_s(X_t, u)\|_2^2 \asymp d^{1-u} \to 0. \tag{68}$$

Since the posterior weights satisfy $0 \leq \gamma_s(X_t, u) \leq 1$, each pairwise contribution to the entropy production is bounded by the collapsing score gap:

$$0 \leq \gamma_s(X_t, u)\lambda_{rs}(u) \leq \lambda_{rs}(u) \to 0. \tag{69}$$

Therefore, for fixed $K$,

$$\frac{\partial}{\partial t} H(Z|X_t) = \frac{g_t^2}{4Kv(u)} \sum_{r=1}^{K} \mathbb{E}\left[ \sum_{s \neq r} \gamma_s(X_t, u)\lambda_{rs}(u) \,\bigg|\, Z = r \right] \to 0. \tag{70}$$

Finally, at $u = 1$, the control parameter satisfies $\lambda_{rs} = O(1)$, posterior competition is no longer exponentially suppressed while the score gaps have not yet collapsed. This is the only asymptotic regime among the three in which the pairwise entropy production terms can remain non-negligible. Therefore, in the VP case, the entropy production concentrates around the scale $t_s(d) = \frac{1}{2}\log d$, which coincides with the speciation time. Consequently, maxima of the full and partitioned class-conditional entropy production provide an operational tool for quantifying the fine-grained and coarse-grained speciation windows, respectively.

## Speciation for Variance-exploding Processes

For variance-exploding EDM-style processes, we have $\alpha_t = 1$, and $\sigma_t^2 = t^2$. The control parameter therefore becomes:

$$v(t) = \sigma_0^2 + t^2, \qquad m_k(t) = \mu_k, \qquad \lambda_{rs}(t) = \frac{\|m_r(t) - m_s(t)\|_2^2}{v(t)} = \frac{\|\mu_r - \mu_s\|_2^2}{\sigma_0^2 + t^2}. \tag{71}$$

Under the scaling assumption $\|\mu_r - \mu_s\|_2^2 \asymp d$, we then obtain the time-dependent bounds for the control parameter along with the speciation timescale by again requiring $\lambda_{rs}(t_s(d)) \asymp 1$:

$$\lambda_{rs}(t) \asymp \frac{d}{\sigma_0^2 + t^2}, \qquad t_s(d) \asymp \sqrt{d}. \tag{72}$$

We can repeat the reasoning from the previous section to study the control parameter under the rescaled time $u := \frac{t}{t_s(d)}$ or more specifically the asymptotic behavior of the mean and standard deviation of $\Lambda_{rs}(X_t, u)$:

$$\lambda_{rs}(u) \asymp \frac{1}{\sigma_0^2/d + u^2} \asymp u^{-2},$$

$$\mathbb{E}\left[\Lambda_{rs}(X_t, u) \,\middle|\, Z = r\right] = \frac{\lambda_{rs}(u)}{2} \asymp u^{-2}, \qquad \mathrm{Var}\left[\Lambda_{rs}(X_t, u) \,\middle|\, Z = r\right] = \lambda_{rs}(u) \asymp u^{-2}. \tag{73}$$

Therefore, unlike in the VP case, the rescaled VE control parameter does not behave like a power of $d$ that diverges on one side of the transition and vanishes on the other. After setting $t = u\sqrt{d}$, we get $\lambda_{rs}(u) \asymp u^{-2}$, so an $O(1)$ range of rescaled times $u$ remains nontrivial as $d \to \infty$. Consequently, the transition does not sharpen into a dimension-independent window on the physical time axis. Instead, because $t = u\sqrt{d}$, an $O(1)$ interval in $u$ corresponds to an $O(\sqrt{d})$ interval in $t$. Thus, VE exhibits a speciation scale $t_s(d) \asymp \sqrt{d}$, but the mixing region broadens on the original time axis rather than remaining of finite additive width as in the VP case.

**TL;DR: Speciation for VP and VE**

Here we provide a short summary of the asymptotic results for both variance-preserving and variance-exploding processes. In both cases, the entropy production is controlled by the pairwise parameter:

$$\frac{\partial}{\partial t} H(Z|X_t) = \frac{g_t^2}{4K\,v(t)} \sum_{r=1}^{K} \int p_t(x|Z=r) \sum_{s \neq r} \gamma_s(x,t) \lambda_{rs}(t)\,\mathrm{d}x,$$

$$\lambda_{rs}(t) = \frac{\|m_r(t) - m_s(t)\|_2^2}{v(t)}, \qquad \gamma_s(X_t, t) = \frac{e^{-\Lambda_{rs}(X_t,t)}}{1 + \sum_{l \neq r} e^{-\Lambda_{rl}(X_t,t)}}$$

(74)

Under source class $Z = r$, the pairwise log-odds satisfy the following:

$$\Lambda_{rs}(X_t, t) \mid Z = r \sim \mathcal{N}\left(\frac{\lambda_{rs}(t)}{2}, \; \lambda_{rs}(t)\right).$$

(75)

Thus, speciation occurs when $\lambda_{rs}(t) \asymp 1$, i.e., posterior competition is non-trivial, but the score gap has not yet collapsed.

*Table 1.* **Compact Speciation Results for VP and VE-EDM Processes (Isotropic Finite Gaussian Mixtures).**

| Quantity | VP | VE-EDM |
|---|---|---|
| **Forward scaling** | $m_k(t) = e^{-t}\mu_k$ | $m_k(t) = \mu_k$ |
| **Variance** | $v(t) = e^{-2t}\sigma_0^2 + 1 - e^{-2t}$ | $v(t) = \sigma_0^2 + t^2$ |
| **Control parameter** | $\lambda_{rs}(t) \asymp d e^{-2t}$ | $\lambda_{rs}(t) \asymp \dfrac{d}{\sigma_0^2 + t^2}$ |
| **Speciation scale** | $t_s(d) = \dfrac{1}{2} \log d + O(1)$ | $t_s(d) \asymp \sqrt{d}$ |
| **Rescaled behavior** | $\lambda_{rs}(u) \asymp d^{1-u}$ | $\lambda_{rs}(u) \asymp u^{-2}$ |
| **Transition behavior** | Sharp at $u = 1$ | Spread over an $O(1)$ range of $u$ |
| **Physical-time window** | $O(1)$ around $\dfrac{1}{2} \log d$ | $O(\sqrt{d})$ |

The key difference is that VP exponentially dampens the class means, which makes the transition sharpen around $u = 1$. For VE instead, the growing variance reduces the SNR only polynomially. Consequently, VE still has a speciation scale, but the mixing region broadens on the original time axis rather than concentrating into a finite additive window. This can be seen in the figure below.

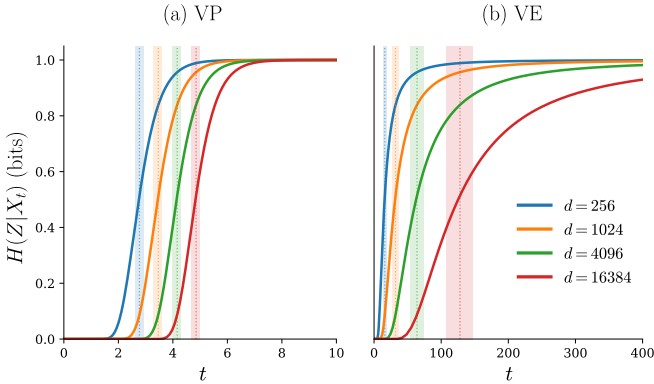

*Figure 5.* **Entropy in VP and VE-EDM.**

## A.4. Extension to General Class Structure

We now return to the VP setting and relate the entropy-production identity (42) to the general speciation framework of Achilli et al. (2026).

We start by introducing the basic notions from Achilli et al. (2026). We assume that the distribution can be written as

$$p_0(a) = \sum_{r=1}^{N} w_r \, P_r(a), \tag{76}$$

Where $P_r$ represents the distribution of candidate classes, while the weights $w_r$ define their prior probabilities.

Since such a decomposition is not unique, Achilli et al. (2026) restrict to decompositions that are operationally meaningful from the viewpoint of Bayesian attribution. More precisely, the decomposition above is called a Proper Density Decomposition if, after adding a small but finite amount of Gaussian noise to a typical sample from class $r$, a Bayes classifier still attributes it to class $r$ with probability tending to one as the dimension grows. Equivalently, for every $r \neq s$, there exists a noise level of order one and a sequence $\epsilon_d \xrightarrow[d\to\infty]{} 0$ such that the posterior probability of attributing a noise-corrupted sample from class $r$ to class $s$ is at most $\epsilon_d$ with high probability.

This notion ensures that the class decomposition is not merely formal, but corresponds to components that remain statistically distinguishable under small finite corruption. In particular, it provides a meaningful notion of class label that can then be followed along the forward diffusion. Furthermore, it implies that the class-conditional entropy is zero at the start of the noising process.

The second key notion is that of a Pure Density Decomposition. This is a Proper Density Decomposition for which the class-conditioned intensive log-likelihoods self-average. Concretely, if a sample is generated from class $r$, then the likelihood under class $s$ admits the large-dimensional form

$$P_s(a) = \exp\big(d\, f_{rs} + o(d)\big), \tag{77}$$

where

$$f_{rs} := \frac{1}{d}\, \mathbb{E}_{a\sim P_r}\big[\log P_s(a)\big] \tag{78}$$

is the corresponding averaged free-entropy density. More generally, after forward diffusion to time $t$, one has the analogous representation

$$P_s(x;t) = \exp\big(d\, f_{rs}(t) + \delta f_{rs}(x,t)\big), \tag{79}$$

where $\delta f_{rs}(x,t) = o(d)$ collects the sample-dependent fluctuations.

Thus, Proper Density Decomposition gives a meaningful notion of class through Bayesian attribution, while Pure Density Decomposition gives the large-dimensional structure needed to study how class attribution breaks down under diffusion. This is precisely the setting in which the speciation criterion of Achilli et al. (2026) is formulated.

In our notation, the posteriors are

$$\gamma_s(x,t) = \frac{w_s P_s(x;t)}{\sum_{u=1}^{N} w_u P_u(x;t)}. \tag{80}$$

To understand when $\mathcal{E}_{r,s}(t)$ can be large, we can write it as

$$\mathcal{E}_{r,s}(t) = \mathbb{E}_{X_t \sim p_t(\cdot|Z=r)}\left[\frac{e^{-\Lambda_{rs}(X_t,u)}}{1+\sum_{n\neq r} e^{-\Lambda_{rn}(X_t,u)}} \, \|s_r(X_t,t) - s_s(X_t,t)\|_2^2\right], \tag{81}$$

where

$$\Lambda_{rs}(x,t) = \log \frac{w_r}{w_s} + \log P_r(x;t) - \log P_s(x;t). \tag{82}$$

Under a Pure Density Decomposition, when $X_t$ is generated from class $r$, one has

$$\Lambda_{rs}(X_t,t) = \log \frac{w_r}{w_s} + d\big(f_{rr}(t) - f_{rs}(t)\big) + \big(\delta f_{rr}(X_t,t) - \delta f_{rs}(X_t,t)\big). \tag{83}$$

This is analogous to the Gaussian decomposition from the previous subsection: the averaged free-entropy difference gives the deterministic class advantage, while the fluctuation difference controls the random spread of the log-odds.

Hence the posterior factor in (9) is directly controlled by the competition between the average free-entropy gap and its fluctuations. In particular, $\gamma_s(X_t, t)$ is exponentially small when $\Lambda_{rs}(X_t, t) \gg 1$, becomes order one when $\Lambda_{rs}(X_t, t) = O(1)$, and remains bounded by 1 at all times.

To understand the behavior of $\Lambda_{rs}$, we need to compare its deterministic drift with its fluctuations. In particular, we want to know when the mean free-entropy gap remains macroscopically positive and when fluctuations can reduce the log-odds to the $O(1)$ regime, where posterior uncertainty becomes non-negligible. Since the speciation time is expected to diverge logarithmically with dimension, it is natural to analyze both quantities through their large-$t$ expansion in powers of $e^{-2t}$.

To describe this regime, define

$$\Delta_{rs}a_i := \langle a_i \rangle_r - \langle a_i \rangle_s, \qquad \Delta_{rs}C_{ij} := \left(\langle a_i a_j \rangle - \langle a_i \rangle \langle a_j \rangle\right)_r - \left(\langle a_i a_j \rangle - \langle a_i \rangle \langle a_j \rangle\right)_s, \tag{84}$$

and

$$A_{rs} := \sum_{i=1}^{d} (\Delta_{rs}a_i)^2, \qquad B_{rs} := \sum_{i,j=1}^{d} (\Delta_{rs}C_{ij})^2. \tag{85}$$

Appendix A of Achilli et al. (2026) shows that, at large times,

$$f_{rr}(t) - f_{rs}(t) \simeq \frac{A_{rs}}{2d}\left(e^{-2t} + e^{-4t}\right) + \frac{B_{rs}}{4d}e^{-4t} + e^{-4t}S_{rs}, \tag{86}$$

while

$$\text{Var}\left[\frac{1}{d}\log P_r(X_t; t) - \frac{1}{d}\log P_s(X_t; t)\right] \simeq \frac{A_{rs}}{d^2}\left(e^{-2t} + 2e^{-4t}\right) + \frac{B_{rs}}{2d^2}e^{-4t}. \tag{87}$$

These formulas make the two possible leading mechanisms transparent. If the classes differ already at the level of their first moments, the $e^{-2t}A_{rs}$ term dominates. If their first moments coincide, the leading contribution comes from the covariance sector and is governed by $e^{-4t}B_{rs}$. This motivates the pairwise scaling variable

$$\lambda_{rs}(t) := \begin{cases} e^{-2t}A_{rs}, & A_{rs} > 0, \\ e^{-4t}B_{rs}, & A_{rs} = 0. \end{cases} \tag{88}$$

The speciation window is precisely the regime in which $\lambda_{rs}(t) = O(1)$. Equivalently, we define the pairwise speciation time $t_{rs}$ by the condition

$$\lambda_{rs}(t_{rs}) = O(1).$$

Next, we inspect the score difference on the same large-$t$ scale. Appendix A of Achilli et al. (2026) gives

$$\frac{1}{d}\log P_r(x; t) - \frac{1}{d}\log P_s(x; t) = \frac{e^{-t}}{d\Delta_t}\sum_i x_i \Delta_{rs}a_i + \frac{e^{-2t}}{2d\Delta_t}\sum_{i,j} x_i \Delta_{rs}C_{ij} x_j + O\big((xe^{-t})^3\big), \tag{89}$$

where $\Delta_t = 1 - e^{-2t}$. Differentiating with respect to $x$ yields

$$s_r(x, t) - s_s(x, t) = \frac{e^{-t}}{\Delta_t}\Delta_{rs}a + \frac{e^{-2t}}{\Delta_t}\Delta_{rs}C\, x + O\big(e^{-3t}\|x\|_2^2\big). \tag{90}$$

Hence, if $A_{rs} > 0$, the leading score-gap scale is $e^{-2t}A_{rs}$, whereas if $A_{rs} = 0$, the leading scale is $e^{-4t}B_{rs}$. In both cases, the same variable $\lambda_{rs}(t)$ controls the decay of the score separation.

### Before Speciation in the large-$t$ Regime

Assume that $t$ is large enough for the expansions above to be valid, but still such that $\lambda_{rs}(t) \to \infty$. Then the deterministic part of $\Lambda_{rs}$ is of order $\lambda_{rs}(t)$, while its fluctuations are only of order $\sqrt{\lambda_{rs}(t)}$. Therefore, the mean still dominates, and

$$\Lambda_{rs}(X_t, t) \to +\infty \tag{91}$$

under source class $r$. It follows that

$$\gamma_s(X_t, t) \leq e^{-\Lambda_{rs}(X_t, t)} \to 0 \tag{92}$$

with high probability. Thus, the posterior uncertainty factor is exponentially suppressed before speciation.

At the same time, the score-gap expansion shows that

$$\|s_r(X_t, t) - s_s(X_t, t)\|_2^2 = O(\lambda_{rs}(t)) \tag{93}$$

in the mean-distinguished case, and analogously after expectation in the covariance-distinguished case under mild moment assumptions. Hence the score gap grows at most polynomially in the same scaling variable that drives the exponential decay of $\gamma_s$. The exponential suppression therefore wins, and

$$\mathcal{E}_{r,s}(t) \to 0 \tag{94}$$

throughout the late-time pre-speciation regime.

## After Speciation and at Sufficiently Late Times

Once $t$ moves beyond the speciation window, the posterior no longer sharply resolves the source class, so $\gamma_s$ need not be small. However, the same score-gap expansion shows that for any fixed pair $(r, s)$,

$$\|s_r(X_t, t) - s_s(X_t, t)\|_2^2 \to 0 \tag{95}$$

at sufficiently late times. Indeed, both the $e^{-t}$ contribution coming from first moments and the $e^{-2t}$ contribution coming from second moments vanish together with higher orders. Since $\gamma_s(X_t, t) \leq 1$, it follows that

$$\mathcal{E}_{r,s}(t) \to 0 \tag{96}$$

at sufficiently late times.

## At Speciation

It remains to understand what happens in the critical window $t = t_{rs} + O(1)$, that is, when $\lambda_{rs}(t) = O(1)$. In this regime the deterministic free-entropy gap and its fluctuations are of the same order, so (83) implies that

$$\Lambda_{rs}(X_t, t) = O(1), \tag{97}$$

and therefore the posterior weight $\gamma_s(X_t, t)$ is no longer exponentially suppressed. This is exactly the point made in Achilli et al. (2026): Bayes attribution between the two classes becomes genuinely uncertain in this window. At the same time, the score-gap expansion shows that the pair $(r, s)$ has not yet been completely washed out by the forward process. Thus the factor $\|s_r - s_s\|_2^2$ is still non-negligible on this scale.

Moreover, the class entropy is asymptotically zero at the beginning of the forward process, while at later times the forward diffusion necessarily produces nontrivial class uncertainty once classes start to mix. Since we have shown that the pairwise contribution vanishes both before the speciation window and again at sufficiently late times, the increase in class uncertainty associated with the onset of $r$–$s$ mixing must be carried by an intermediate time interval. Therefore, by the same reasoning as in the Gaussian case, the natural location of the pairwise entropy-production peak is the speciation window itself:

$$t_{\text{peak}, rs} = t_{rs} + O(1). \tag{98}$$

Combining the three regimes gives the same picture as in the Gaussian analysis. Before speciation, the posterior uncertainty factor is exponentially suppressed. At speciation, the log-odds enter the $O(1)$ regime and the posterior uncertainty becomes nontrivial while the score gap is still non-negligible. At sufficiently late times, the score fields themselves collapse together. Hence, the directed pairwise entropy-production contribution is expected to develop a peak in the same critical window as the pairwise speciation time $t_{rs}$. In this sense, the peak mechanism in our entropy-production formula is the same one identified in Achilli et al. (2026). Examples of this behavior are shown in Figure 6.

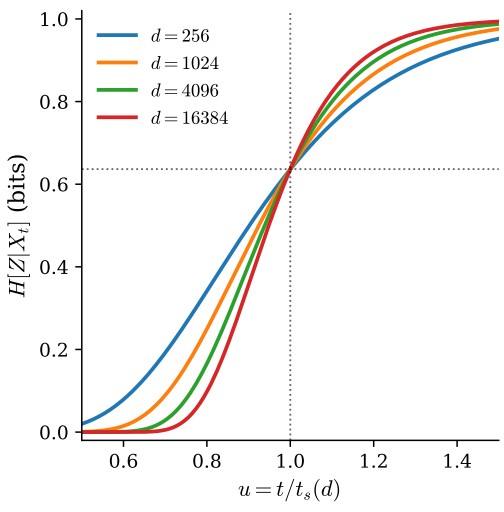

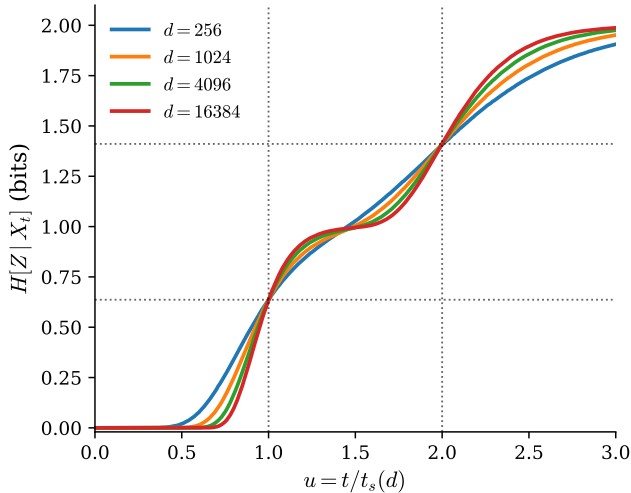

*(a)* Mixture of two concentric Gaussians

*(b)* Hierarchical mixture of four Gaussians

*Figure 6.* Class-conditional entropy $H(Z|X_t)$ for Gaussian mixtures under the VP forward process, shown as a function of the rescaled time $u = t/t_s(d)$ with $t_s(d) = \frac{1}{4}(\log d - 1)$. Panel (a) corresponds to a mixture of two zero-mean Gaussians with covariance matrices $(1 \pm 0.7)I_d$. Panel (b) corresponds to a four-Gaussian hierarchical mixture, consisting of two pairs with opposite means and distinct covariances within each pair. In this case the entropy displays a two-step increase: the first rise at $u = 1$ is associated with merging within equal-mean pairs, while the second rise at $u = 2$ corresponds to merging between the two mean-separated pairs.

## B. Implementation Details and Estimator Robustness

In this section, we further discuss implementation choices and analyze the estimator's robustness in the setting of (an)isotropic Gaussian mixture models. As stated in Section 4.5, the class-conditional entropy and entropy production are insensitive to individual competition between classes. We therefore proposed partitioned class-conditional entropy and entropy production as diagnostics to probe speciation at different levels of abstraction. For reference, both identities, written in their expanded form, along with auxiliary definitions, are provided below again:

$$H(Y_{\mathcal{S}}|X_t) = -\sum_{i \in \{0,1\}} \pi_i \int q_{i,\mathcal{S}}(x,t) \sum_{y \in \{0,1\}} \gamma_{y,\mathcal{S}}^{\pi}(x,t) \log \gamma_{y,\mathcal{S}}^{\pi}(x,t) \, \mathrm{d}x,$$

$$\frac{\partial}{\partial t} H(Y_{\mathcal{S}}|X_t) = \frac{g_t^2}{2} \sum_{i \in \{0,1\}} \pi_i \int q_{i,\mathcal{S}}(x,t) \left(\gamma_{1-i,\mathcal{S}}^{\pi}(x,t)\right)^2 \|s_{0,\mathcal{S}}(x,t) - s_{1,\mathcal{S}}(x,t)\|_2^2 \, \mathrm{d}x, \tag{99}$$

with

$$q_{i,\mathcal{S}}(x,t) := p_t(x|Y_{\mathcal{S}} = i) := \sum_{k \in \mathcal{S}_i} p_t(x|Z = k) \frac{w_k}{\sum_{j \in \mathcal{S}_i} w_j}, \qquad i \in \{0,1\},$$

$$p_{t,\mathcal{S}}^{\pi}(x) := \pi_0 q_{0,\mathcal{S}}(x,t) + \pi_1 q_{1,\mathcal{S}}(x,t), \qquad \gamma_{i,\mathcal{S}}^{\pi}(x,t) := \frac{\pi_i q_{i,\mathcal{S}}(x,t)}{p_{t,\mathcal{S}}^{\pi}(x)}, \qquad i \in \{0,1\},$$

$$s_{i,\mathcal{S}}(x,t) := \nabla_x \log q_{i,\mathcal{S}}(x,t), \qquad s_{\mathrm{mix},\mathcal{S}}^{\pi}(x,t) := \nabla_x \log p_{t,\mathcal{S}}^{\pi}(x) = \sum_{i \in \{0,1\}} \gamma_{i,\mathcal{S}}^{\pi}(x,t) s_{i,\mathcal{S}}(x,t), \tag{100}$$

$$\pi_0 := \frac{\sum_{k \in \mathcal{S}_0} w_k}{\sum_{j \in \mathcal{S}} w_j}, \qquad \pi_1 := 1 - \pi_0, \qquad w_k := p(Z = k).$$

**Partition priors** The first step in estimating the entropy or its production form is to choose a partition $(\mathcal{S}_0, \mathcal{S}_1)$, with $\mathcal{S}_0 \cup \mathcal{S}_1 = S \subseteq \{1, ..., K\}$. Accordingly, the priors are set as follows:

$$\pi_0 = \frac{\sum_{k \in \mathcal{S}_0} w_k}{\sum_{j \in \mathcal{S}} w_j}, \qquad \pi_1 = 1 - \pi_0. \tag{101}$$

We can see that for equiprobable class structures, partitions with $|\mathcal{S}_0| = |\mathcal{S}_1|$ have partition priors of 0.5. For class-classes comparisons, the partition priors can become heavily one-sided. In our experiments on EDM2-XS, we compare classes against their complements $\left(\mathcal{S}_0 = \{k\}, \mathcal{S}_1 = \{j\}_{j \neq k}^{K}\right)$, which have $\pi_0 = 0.001$ and $\pi_1 = 0.999$ (ImageNet contains 1K classes). Consequently, the summands multiplied by the specific class prior carry negligible weight on the entropy and its production. In settings with an unbounded class set (e.g., T2I), the class specific summand would completely vanish. To combat this issue, we choose to treat both sides of any partition as equally probable, which does not compromise its quality as a speciation observable (Appendix A.3). Reweighing priors accordingly also balances the posteriors:

$$\pi^{\mathrm{bal}} := (\pi_0 = \pi_1 = 0.5),$$

$$\gamma_{i,\mathcal{S}}^{\pi^{\mathrm{bal}}}(x,t) = \frac{\pi^{\mathrm{bal}} q_{i,\mathcal{S}}(x,t)}{\pi^{\mathrm{bal}} q_{0,\mathcal{S}}(x,t) + \pi^{\mathrm{bal}} q_{1,\mathcal{S}}(x,t)} = \frac{\frac{|\mathcal{S}_{1-i}|}{|\mathcal{S}_i|} \sum_{k \in \mathcal{S}_i} p_t(x|Z = k)}{\frac{|\mathcal{S}_{1-i}|}{|\mathcal{S}_i|} \sum_{k \in \mathcal{S}_i} p_t(x|Z = k) + \sum_{j \in \mathcal{S}_{1-i}} p_t(x|Z = j)}, \tag{102}$$

which becomes the following for class-against-complement partitions:

$$\gamma_{0,\mathcal{S}}^{\pi^{\mathrm{bal}}}(x,t) = \frac{p_t(x|Z = k)}{p_t(x|Z = k) + p_t(x|Z \neq k)} = \frac{p_t(Z = k|x)}{p_t(Z = k|x) + \frac{1}{K-1} p_t(Z \neq k|x)}. \tag{103}$$

Consequently, under the balanced prior, the partitioned class-conditional entropy can be expressed through a Jensen-Shannon Divergence between the corresponding partitions:

$$H_{\mathrm{bal}}(Y_{\mathcal{S}}|X_t) = \log 2 - \mathrm{JSD}(q_{0,\mathcal{S}}||q_{1,\mathcal{S}}). \tag{104}$$

**Comparisons with the complement** In class-complement partitions, the accessibility of the complement model decreases with the complexity of the class structure. We previously mentioned the T2I case, where the set of classes is essentially infinitely large. However, at the same time its difference to the full model shrinks. Therefore, we choose to use the full or unconditional model as an approximation in those cases instead. Pairing this with the balanced priors, we obtain the posterior from above, just for the adapted partition $(\mathcal{S}_0 = \{k\}, \mathcal{S}_1 = \{j\}_{j=1}^K)$:

$$\gamma_{0,\mathcal{S}}^{\pi^{\mathrm{bal}}}(x,t) = \frac{p_t(x|Z=k)}{p_t(x|Z=k) + p_t(x)} = \frac{p_t(Z=k|x)}{\frac{K+1}{K}p_t(Z=k|x) + \frac{1}{K}p_t(Z\neq k|x)}. \tag{105}$$

We can see that as $K$ grows, the two quantities converge. However, the approximation also enters through the expectation, which requires that both mixtures (class-complement, class-unconditional), that is:

$$p_{t,\mathcal{S}}^{\pi^{\mathrm{bal}}}(x) = \pi^{\mathrm{bal}}p_t(x|Z=k) + \pi^{\mathrm{bal}}p_t(x|Z\neq k), \qquad p_{t,\mathcal{S}}^{\pi^{\mathrm{bal}}}(x) = \pi^{\mathrm{bal}}p_t(x|Z=k) + \pi^{\mathrm{bal}}p_t(x), \tag{106}$$

produce approximately equivalent samples during Monte Carlo sampling. This is the case for models trained on moderately sized class structures, such as the EDM2 family, but does not hold for the early T2I diffusion models.

**Estimating the posterior** Both entropy and its production form require access to the posteriors, which we estimate using the iterative update procedure in reverse time from (Koulischer et al., 2025a) defined for stochastic ancestral/DDPM-style sampling:

$$\begin{aligned}
\log p(Z=k|x_{t:T}) = \log p(Z=k|x_t) &= \log p(Z=k) + \log\left(\frac{p(x_{t:T}|Z=k)}{p(x_{t:T})}\right) \\
&= \log p(Z=k) + \sum_{\tau=t+1}^{T} \log\left(\frac{p_\theta(x_{\tau-1}|x_\tau, Z=k)}{p_\theta(x_{\tau-1}|x_\tau)}\right) \\
&= \log p(Z=k|x_{t+1}) + \log\left(\frac{p_\theta(x_t|x_{t+1}, Z=k)}{p_\theta(x_t|x_{t+1})}\right),
\end{aligned} \tag{107}$$

where the first and last equality follow from the Markov independence, i.e., the least noisy state contains all the information from the previous states. We write the update in log-odds form and obtain the posteriors according to:

$$\begin{aligned}
\Lambda_{0,\mathcal{S}}(x,t) &:= \log\left(\frac{\gamma_{0,\mathcal{S}}^\pi(x,t)}{\gamma_{1,\mathcal{S}}^\pi(x,t)}\right) = \log\left(\frac{\gamma_{0,\mathcal{S}}^\pi(x,t+1)}{\gamma_{1,\mathcal{S}}^\pi(x,t+1)}\right) + \log\left(\frac{p_\theta(x_t|x_{t+1}, Y_\mathcal{S}=0)}{p_\theta(x_t|x_{t+1}, Y_\mathcal{S}=1)}\right), \\
\gamma_{0,\mathcal{S}}^\pi(x,t) &= \frac{\exp\left(\Lambda_{0,\mathcal{S}}(x,t)\right)}{\exp(\Lambda_{0,\mathcal{S}}(x,t)) + 1}, \qquad \gamma_{1,\mathcal{S}}^\pi(x,t) = 1 - \gamma_{0,\mathcal{S}}^\pi(x,t),
\end{aligned} \tag{108}$$

which in continuous time becomes the Girsanov likelihood ratio between the two conditional reverse SDEs:

$$\mathrm{d}X_t = \left(f(X_t,t) - g_t^2 s_{i,\mathcal{S}}(X_t,t)\right)\mathrm{d}t + g_t \mathrm{d}\widetilde{W}_t, \qquad i \in \{0,1\}, \qquad \mathrm{d}t < 0, \tag{109}$$

$$\begin{aligned}
\mathrm{d}\Lambda_{0,\mathcal{S}}(X_t,t) = {}& \left(s_{0,\mathcal{S}}(X_t,t) - s_{1,\mathcal{S}}(X_t,t)\right)^\top \mathrm{d}X_t \\
& + \left[-f(X_t,t)^\top\left(s_{0,\mathcal{S}}(X_t,t) - s_{1,\mathcal{S}}(X_t,t)\right) + \frac{g_t^2}{2}\left(\|s_{0,\mathcal{S}}(X_t,t)\|_2^2 - \|s_{1,\mathcal{S}}(X_t,t)\|_2^2\right)\right]\mathrm{d}t.
\end{aligned} \tag{110}$$

**Estimator Robustness** As mentioned above, these practical choices do not invalidate Algorithm 1 as a speciation observable in the asymptotic limit. To quantify the practical effect of our adaptations in the class-classes settings and in particular the class-complement setting depicted in Figure 1, we studied the bias of Algorithm 1 using the decomposition:

$$\begin{aligned}
H(Y_\mathcal{S}|X_t) - H_{\mathrm{bal},\emptyset,\tilde{p}}(Y_\mathcal{S}|X_t) = {}& H(Y_\mathcal{S}|X_t) - H_{\mathrm{bal}}(Y_\mathcal{S}|X_t) \\
& + H_{\mathrm{bal}}(Y_\mathcal{S}|X_t) - H_{\mathrm{bal},\emptyset}(Y_\mathcal{S}|X_t) \\
& + H_{\mathrm{bal},\emptyset}(Y_\mathcal{S}|X_t) - H_{\mathrm{bal},\emptyset,\tilde{p}}(Y_\mathcal{S}|X_t),
\end{aligned} \tag{111}$$

where the subscripts $\mathrm{bal}$, $\emptyset$, and $\tilde{p}$ denote the use of balanced priors, the unconditional model instead of the true complement, and the posterior approximation, respectively. We use two Gaussian mixture settings (VP, VE-EDM) with 200 components

(classes). We additionally increase the dimension of the distribution while adjusting the component distances according to the scaling from section 4.2. Lastly, we study the effects in the approximation limit. That is, using a number of samples sufficient for accurate numerical integration and posterior approximation.

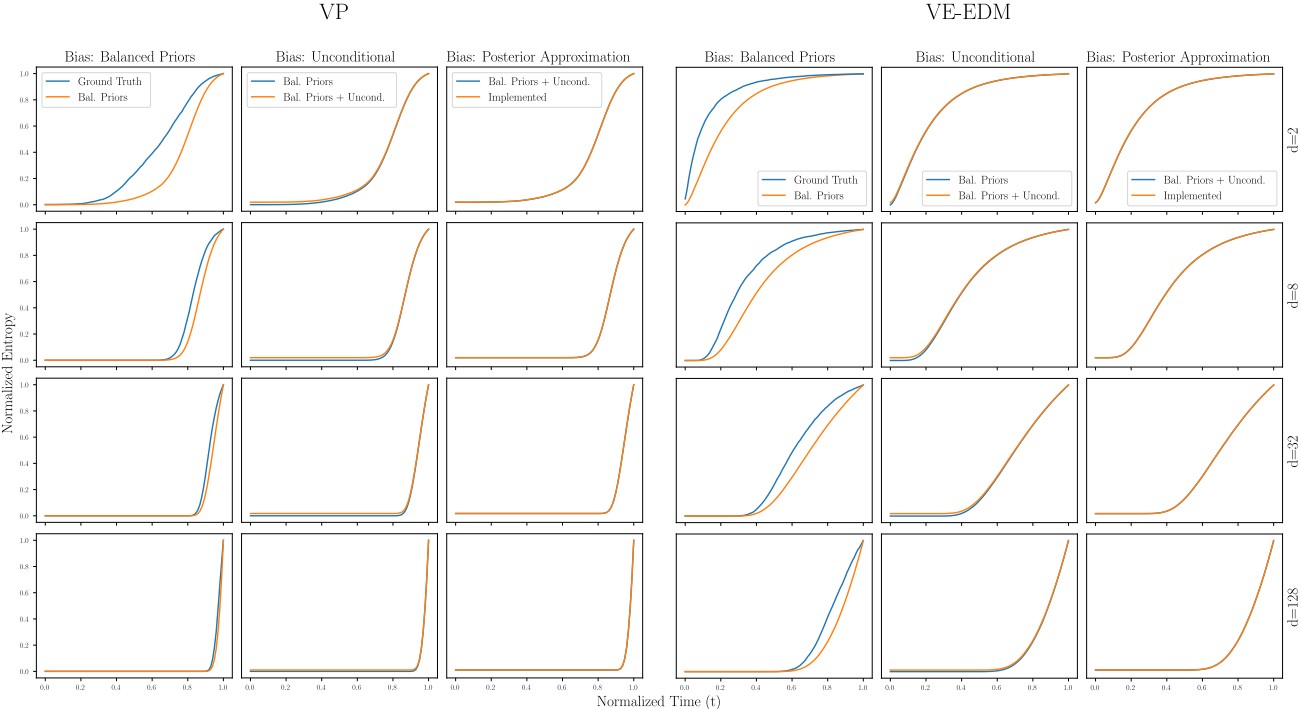

*Figure 7.* **Bias decomposition of Algorithm 1 on VE-EDM/VP Gaussian mixtures with 200 classes**. Columns depict the contributions of the individual terms of Eq. 111 for VP (left) and VE-EDM (right) as dimension increases (i.e., 2, 8, 32, 128). The partition measured here is that of one randomly selected class against its complement. Entropies are normalized to allow for direct comparison.

In our experiments, the strongest bias comes from re-balancing partition priors. But even that effect starts to vanish as the dimensionality of the data distribution increases. Additionally, it does not affect the time of entropy collapse. The bias of the unconditional model remains negligible in the settings studied, which extends to practical settings under the assumption that the unconditional model is not a biased sampler of individual classes (as discussed before). The posterior approximation is exact for a sufficiently large number of integration steps and samples (i.e, 100 and 50K). We furthermore studied the posterior approximation and the re-balancing of priors in a practical setting (i.e., EDM2-XS):

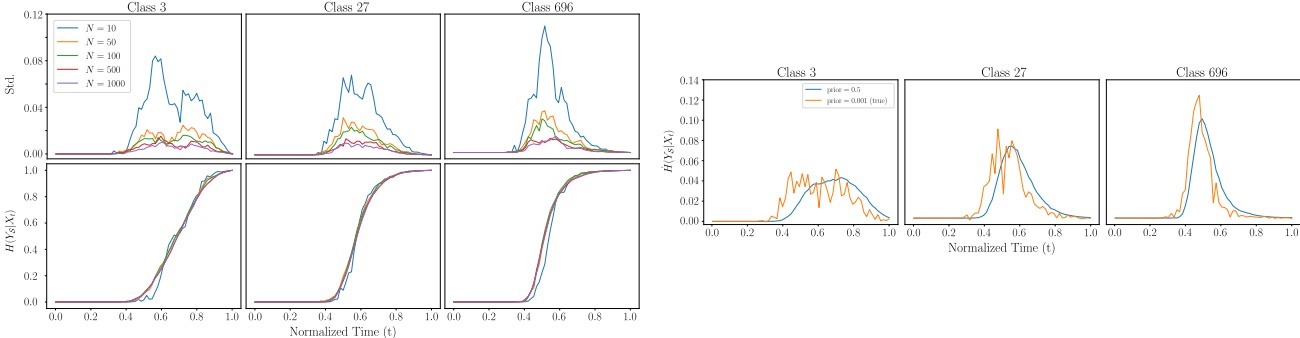

*Figure 8.* **Convergence Analysis of Algorithm 1 on EDM2-XS in class-complement settings**. Left) Convergence of posterior approximation. Each standard deviation is computed using 20 seeds. Right) Difference between natural and balanced priors. For the right panel we used 25K samples. The number of integration steps is fixed at 200. Entropy production is normalized.

Entropy profiles converge using a moderate number of samples. The prior re-balancing does not strongly affect the

characteristic window of the profiles. However, the shape of the profile is altered, which should be considered for potential downstream applications of our algorithm.

**Entropy or entropy production?**    Both entropy and entropy production are equally valid as speciation observables. In our experiments, we estimated the entropy and, after smoothing with a Gaussian kernel, computed the derivative by linearly interpolating. We found that the approximation of the derivative has a higher variance which makes smoothing as a post processing step more difficult.

**Using guidance**    In our guidance experiments, which are shown in Figure 3, the posterior approximation is carried out on the guided trajectories. Consequently, the expectation in Eq. 26 is taken with respect to the guided mixture distribution. The quantity hence estimated is a cross entropy. Note that the posterior approximation remains valid as the increments are still Markovian.

## C. Experimental Details & Additional Results

**Model settings EDM2-XS**    In experiments using the EDM2-XS model trained on ImageNet-512 (Deng et al., 2009), that is, Figure 1 and Figure 3, samples were generated with a stochastic DDIM sampler (Song et al., 2021a) (NFE=64) discretizing uniformly in time according to the EDM schedule with ($\sigma_{\min} = 0.002, \sigma_{\max} = 80.0, \rho = 7.0$) (Karras et al., 2022; Deng et al., 2009). The highlighted entropy profiles for both figures (i.e., *tiger shark*, *eft*, *paintbrush*) were approximated from 3K samples each, while for the remaining class-complement entropy profiles, we used 300 samples, which we selected according to Figure 8. Figure 3 relies on optimized guidance intervals, which we determined using a grid search on ($\sigma_{\text{low}}, \sigma_{\text{high}}, \omega$). Metrics were computed by generating 50K samples in each setting using a stochastic DDIM sampler paired with the mentioned EDM schedule (NFE=64). The metrics used for the evaluation were FID (Heusel et al., 2017), $\text{FD}_{\text{DINOv2}}$ (Stein et al., 2023), precision & recall (Kynkäänniemi et al., 2019; Sajjadi et al.), and a logit classification margin based on Inception-v3. FID and $\text{FD}_{\text{DINOv2}}$ both compute the *Frechet distance* using mean and covariance estimators for real and generated image embeddings obtained from Inception-v3 (Szegedy et al., 2016) and DINOv2 (Oquab et al., 2024), respectively. Precision and recall measure the percentage of images generated that are within the data manifold and the percentage of real images that are within the generation manifold, respectively. For this purpose, we used the DINOv2 embeddings and $k = 5$ as the neighborhood size, i.e., the local manifold measure. The data manifold was approximated using 10K samples from the ImageNet-512 test set. The Inc.-v3 logit classification margin was computed by averaging the per-sample logit margins between the generated samples' classes and their nearest neighbors. That is, $\text{logit}_c(x) - \max_{i \neq c} \text{logit}_i(x)$. We optimized two guidance settings: Limited- and full-interval guidance. The metrics used for optimization were FID and $\text{FD}_{\text{DINOv2}}$. The results are reported in the following two tables:

| Objective | $[\sigma_{\text{low}}, \sigma_{\text{high}}], \omega$ | FID ($\downarrow$) | $\text{FD}_{\text{DINOv2}}$ ($\downarrow$) | Inception-v3 ($\uparrow$) | Prec. ($\uparrow$) | Rec. ($\uparrow$) |
|---|---|---|---|---|---|---|
| FID | $[0.27, 1.50], 2.0$ | 3.55 | 126.8 | 3.85 | 0.889 | 0.752 |
| $\text{FD}_{\text{DINOv2}}$ | $[0.93, 5.28], 2.5$ | 7.20 | 87.8 | 4.57 | 0.940 | 0.723 |

*Table 2.* Optimal limited interval guidance settings for FID and $\text{FD}_{\text{DINOv2}}$, along with independent evaluation metrics.

| Objective | $\omega$ | FID ($\downarrow$) | $\text{FD}_{\text{DINOv2}}$ ($\downarrow$) | Inception-v3 ($\uparrow$) | Prec. ($\uparrow$) | Rec. ($\uparrow$) |
|---|---|---|---|---|---|---|
| FID | 0.375 | 4.96 | 131.65 | 3.52 | 0.882 | 0.752 |
| $\text{FD}_{\text{DINOv2}}$ | 1.25 | 7.42 | 101.15 | 4.59 | 0.927 | 0.686 |

*Table 3.* Optimal full guidance settings for FID and $\text{FD}_{\text{DINOv2}}$, along with independent evaluation metrics.

We excluded the full-interval guidance setting that was optimized for FID from Figure 3 as its effects only differ in strength from the other full-interval guidance condition. To provide the reader with a feeling for the visual quality and diversity of the images generated under the other three guidance conditions, we provided images for the classes highlighted in Figure 3 along with images generated by using non-guided conditional generation.

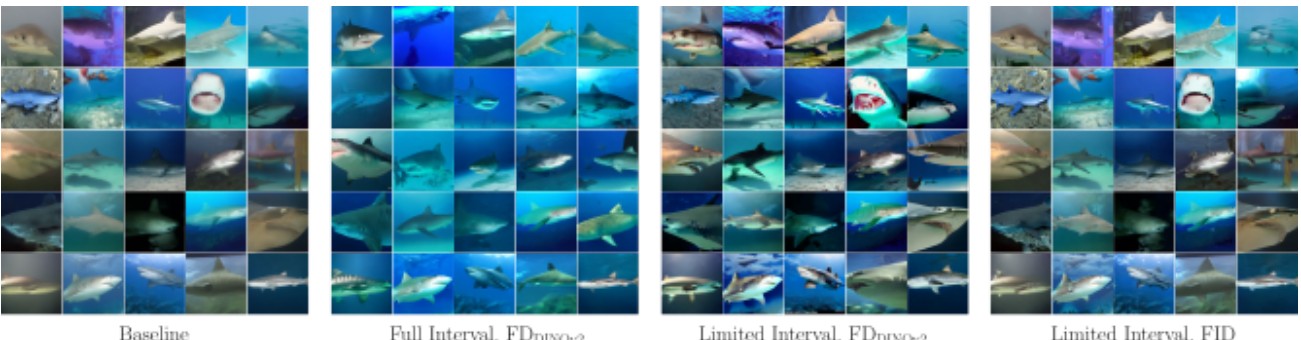

*Figure 9.* **Generation examples for the class tiger shark (3).**

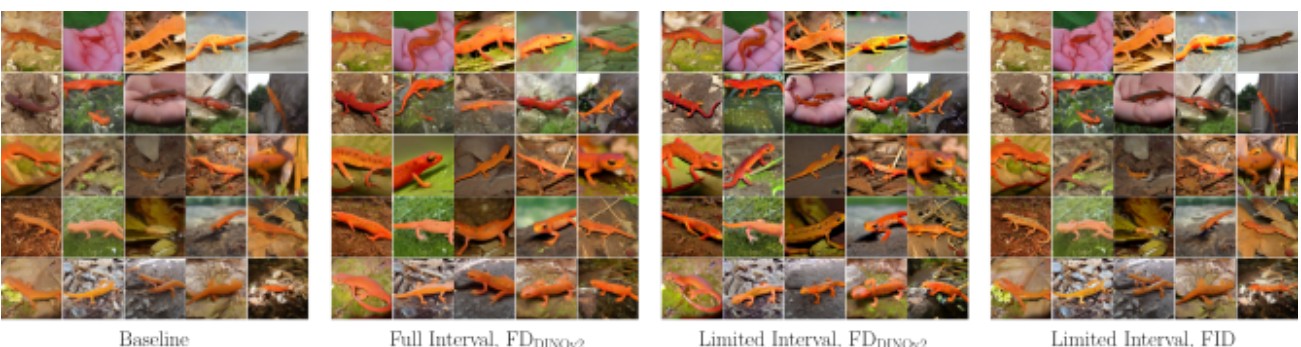

*Figure 10.* **Generation examples for the class eft (27).**

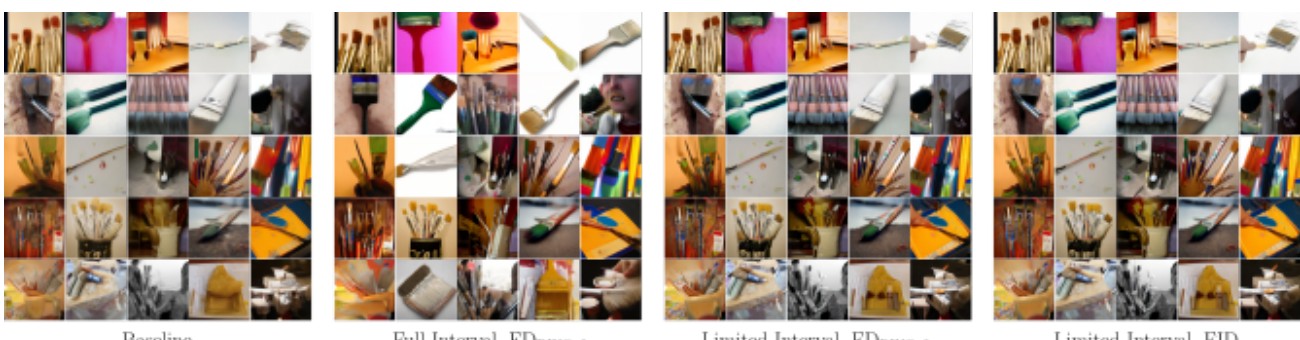

*Figure 11.* **Generation examples for the class paintbrush (696).**

Similarly, we try to complement Figure 3 by relating the highlighted entropy production profiles to guided noiseless predictions for a seeded sample trajectories of each class for each condition. This is shown in Figure 12 below.

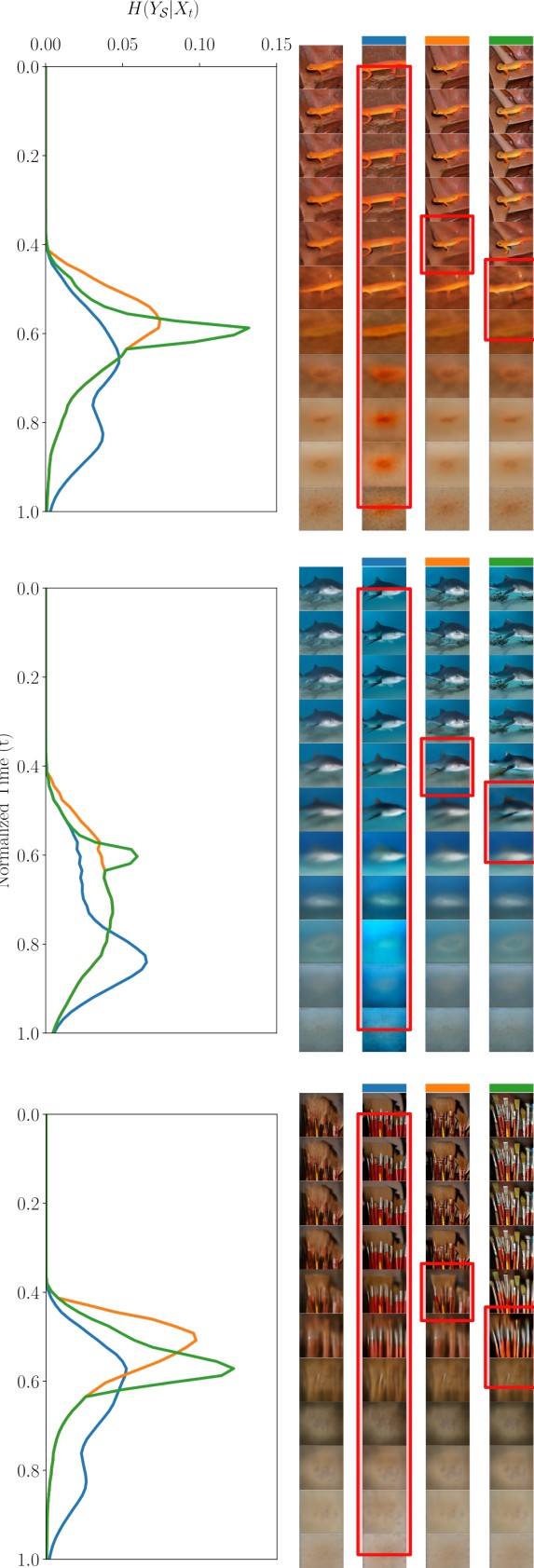

*Figure 12.* **Distortion effects of guidance for the three highlighted classes from Figure 3.** Guidance is applied in the regions framed in red. Entropy production profiles are color matched with denoised predictions.

**Model settings SD 1.5** In experiments involving Stable Diffusion 1.5. (Rombach et al., 2022) trained on LAION5B (Schuhmann et al., 2022), i.e., Figure 4, samples were generated using a stochastic DDIM sampler (Song et al., 2021a) (NFE=100) with a standard DDPM scheduler (Ho et al., 2020). The entropy profiles were generated using 400 samples each. However, we observed visual convergence from around 200 samples on the tested prompts.

This section provides additional entropy profiles together with the corresponding qualitative samples, following the presentation used in Figure 4. The examples serve two purposes. First, they show that the temporal ordering observed in the main text is reproducible across additional samples and base prompts. Second, they illustrate limitations of the metric in cases where the semantic comparison is not cleanly controlled.

A first limitation occurs when the model does not adhere sufficiently to the prompt. In such cases, the measured entropy profile may no longer correspond to the intended semantic attribute, because the attribute is absent or only weakly realized in the generated sample. This failure mode is illustrated in Figure 13, where the cat is missing from the top-left sample. A second limitation arises when the variant prompt differs substantially from the base prompt beyond the targeted attribute. In that case, the conditional entropy does not exclusively measure when the specified attribute is generated; it also reflects broader distributional differences between the base-prompt and variant-prompt samples.

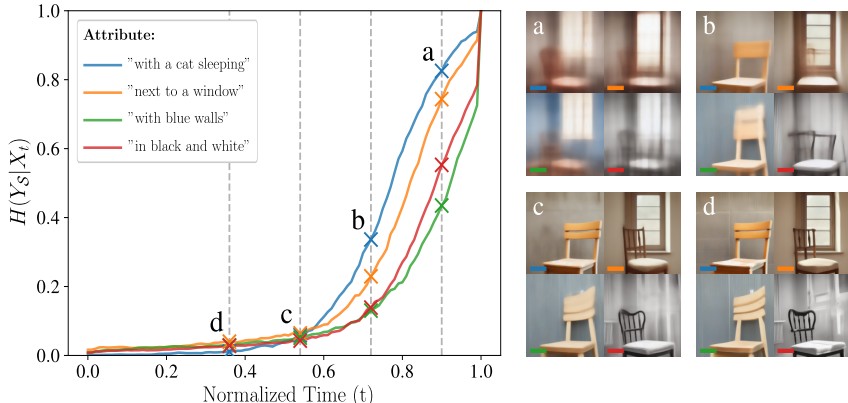

*Figure 13.* **Failure case caused by poor prompt adherence in Stable Diffusion 1.5.** Entropy profiles for binary partitions of the form 'A wooden chair" versus 'A wooden chair + attribute". The failure is visible in the absence of the cat in the top-left sample, indicating that the intended semantic attribute is not reliably generated.

We first consider two additional samples generated from the same base and variant prompts as in the main text. Figures 14 and 15 show that the qualitative sequence of generation remains consistent with the examples reported in the main body. At snapshot (a), only coarse, low-frequency prompt components appear to affect the sample. At snapshot (b), larger structural elements such as windows are resolved. By snapshot (c), the relevant prompt variants are visually expressed in the generated samples.

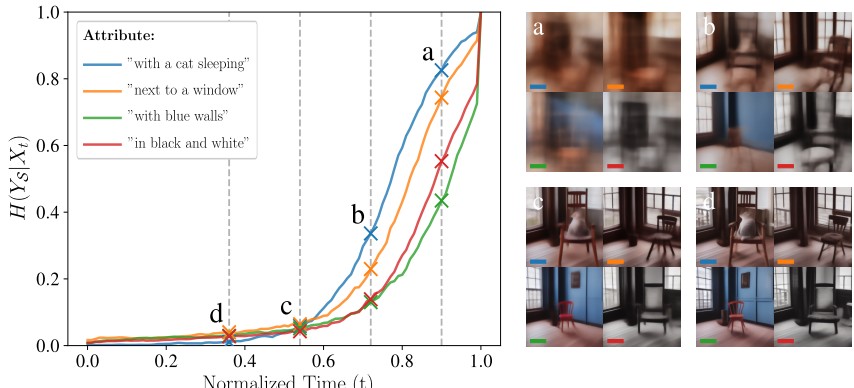

*Figure 14.* **Additional entropy profiles for wooden-chair prompts.** Entropy profiles for binary partitions of the form 'A wooden chair" versus 'A wooden chair + attribute". The temporal ordering of attribute emergence matches the pattern observed in the main text.

The same progression is visible in a second independent sample with the same prompt family. Again, coarse scene-level properties are resolved earlier, whereas more localized or semantically specific attributes emerge later in the sampling trajectory.

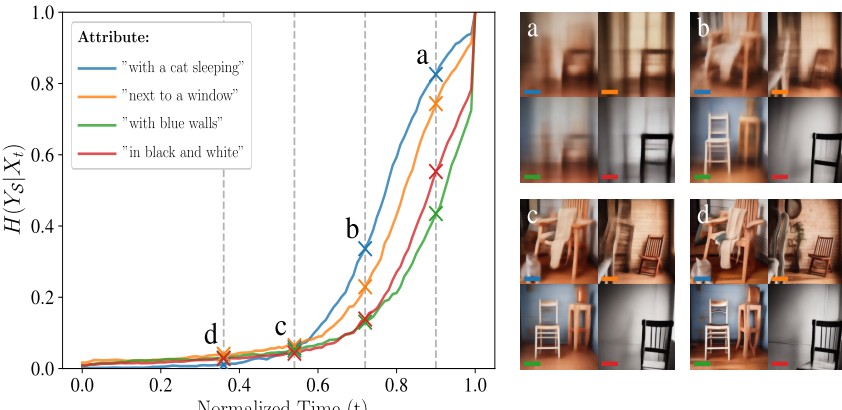

*Figure 15.* **Second additional entropy profile for wooden-chair prompts.** Entropy profiles for binary partitions of the form 'A wooden chair" versus 'A wooden chair + attribute". The profile confirms the same coarse-to-fine ordering of semantic resolution.

Next, we evaluate whether the same qualitative behavior appears for different base prompts. For this purpose, we use the prompts *'A sports car from the front"* and *'A portrait of an orange cat"* along with their, respectively, added attributes. The corresponding results are shown in Figures 16 and 17. Because Stable Diffusion 1.5 exhibits imperfect prompt adherence, prompts must be selected carefully for this type of fine-grained semantic analysis. We therefore manually verified that the chosen prompts were followed sufficiently well by the model. We expect this selection step to become substantially easier for models with stronger prompt adherence.

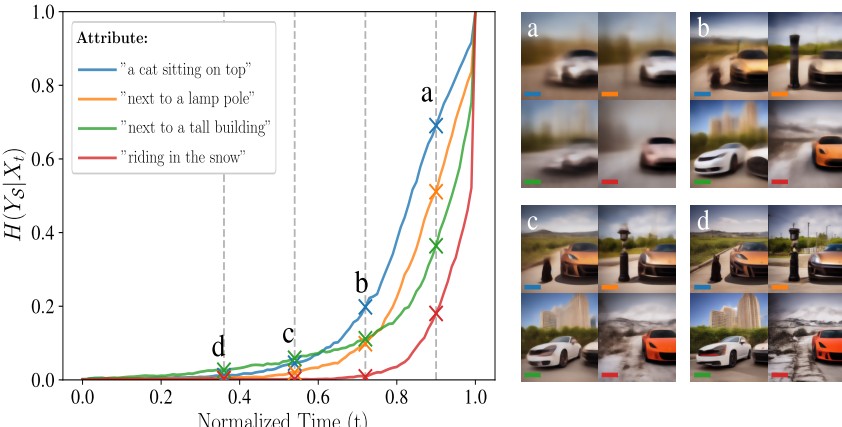

*Figure 16.* **Entropy profiles for sports-car prompts.** Entropy profiles for binary partitions of the form 'A sports car from the front" versus 'A sports car from the front + attribute". Coarse visual changes are resolved earlier than smaller localized attributes.

For both alternative base prompts, we observe trends similar to those in the wooden-chair examples. Low-frequency or global attributes, such as *'in the snow"* or *'in black and white"*, tend to be generated earlier. Larger objects or scene-level modifications, such as *'next to a building"*, also appear relatively early. By contrast, smaller and more localized attributes, such as *'wearing a hat"* or *"a cat sitting on top"*, are resolved later. Overall, the qualitative samples and the corresponding entropy curves remain well aligned.

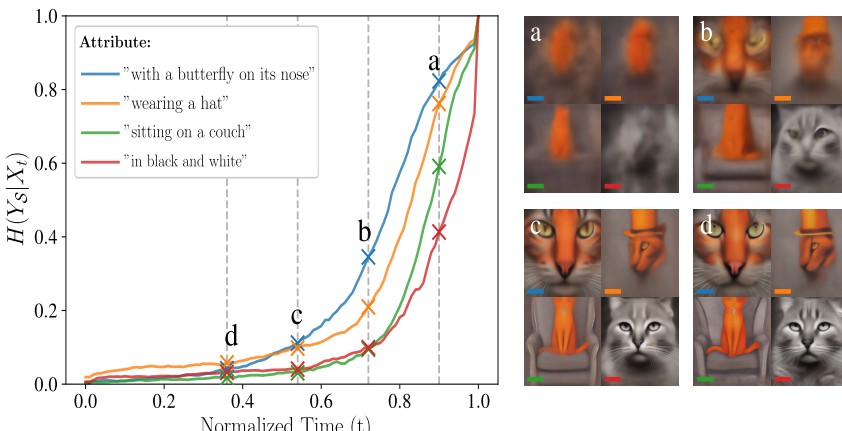

*Figure 17.* **Entropy profiles for orange-cat portrait prompts.** Entropy profiles for binary partitions of the form 'A portrait of an orange cat" versus 'A portrait of an orange cat + attribute". The results again show that global visual changes emerge earlier than smaller object-level or localized semantic modifications.

Finally, we repeat the wooden-chair experiment from the main text with the stronger SD XL base 1.0 model. As shown in Figure 18, the resulting entropy profiles are smoother than for SD 1.5, and the generated samples exhibit better alignment with the corresponding prompts. This is consistent with our interpretation that the metric becomes more reliable when prompt adherence improves: the entropy profile then more cleanly tracks the emergence of the targeted semantic attribute rather than artifacts caused by failed generation or unintended prompt-induced distribution shifts.

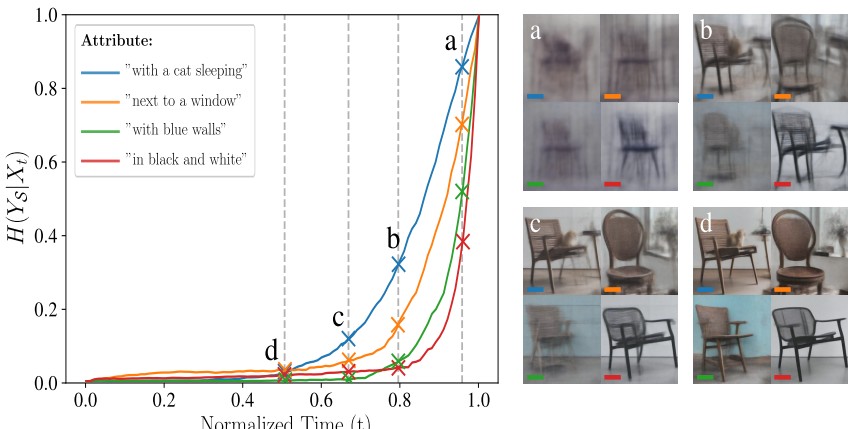

*Figure 18.* **Entropy profiles for the wooden-chair experiment using SD XL base 1.0.** Compared with the corresponding SD 1.5 examples, the entropy profiles are smoother and the generated samples align more reliably with the intended prompt variants. This supports the robustness of the qualitative coarse-to-fine ordering observed in the main text, while also indicating that stronger prompt adherence improves the interpretability of the entropy-based diagnostic.

**Used resources**  The experiments with EDM2-XS were run on a NVIDIA RTX 4090 24 Gb GPU. Approximating a partitioned class-conditional entropy profile for an arbitrary partition on such a node (300 samples, 64 NFEs, stochastic DDIM, batch size: 300) takes approximately 1 min. The Stable Diffusion experiments were run on a NVIDIA H200 NVL 141 Gb GPU. Approximating a partitioned class-conditional entropy profile for an arbitrary partition on such a node (400 samples, 100 NFEs, stochastic DDIM, batch size: 50) takes approximately 0.5 h for the SD 1.5 base model and 4.0 h for SDXL 1.0 base.

