# OpenReview forum: "The Entropic Signature of Class Speciation in Diffusion Models"
_ICML.cc/2026/Conference — ICML 2026 regular_

### Official Review · Reviewer_j2xx · 2026-03-12

**Soundness:** 2
**Presentation:** 3
**Significance:** 2
**Originality:** 3
**Overall Recommendation:** 4
**Confidence:** 4

**Summary:**

The paper proposes estimating the class-conditional entropy given the noisy state to capture the phase transition of diffusion models. Based on the Gaussian mixture assumption, the authors propose practical estimators to estimate the class-conditional entropy using the datasets and the pretrained diffusion model. Experiments on EDM2-XS and Stable Diffusion 1.5 models show that the class-conditional entropy consistently isolates the noise regimes critical for semantic structure formation.

**Compliance With Llm Reviewing Policy:**

Affirmed.

**Final Justification:**

My concerns have been adequately addressed in the authors' rebuttal. I think the weak acceptance rating is appropriate for the paper.

**Key Questions For Authors:**

- In text-to-image models, prompts often carry multiple semantic factors. Could the proposed framework scale to more complex semantic decompositions?
- Could the author extend the empirical results to other diffusion formulations, such as Flow Matching and recent advanced text-to-image diffusion models, such as FLUX and Stable Diffusion 3.5?
- Could the author give a detailed analysis of the bias or error of the proposed estimator?

**Limitations:**

Yes.

**Strengths And Weaknesses:**

### Strength
- The paper is clearly written, and the proposed method is well motivated. The idea of using the conditional entropy to investigate how semantic structure emerges during diffusion sampling is quite interesting.
- The paper proposes a principled method to analyze the semantic structure emergence, which is studied empirically in previous works.

### Weakness
- The conditional entropy estimator in Algorithm 1 relies on a conditional and unconditional noise prediction model. The approximation error of these models may hurt the accuracy of the estimator. The error of the estimator does not have a rigorous theoretical discussion.

- The theoretical and practical cost of the conditional entropy estimation. A detailed analysis of the computational complexity and overhead of the proposed estimator may further strengthen the effectiveness of the proposed estimator.

---

> ### Author Rebuttal · Authors · 2026-03-31
>
> Thank you for reviewing our manuscript and providing helpful feedback and questions.
>
> **Response to**:
>
> - *"[...] Could the proposed framework scale to more complex semantic decompositions?"*
>
> Our framework is not restricted to simple prompt pairs. In principle, one can define binary partitions between two complex prompts, or between one prompt and a reference set, and estimate the corresponding partitioned class-conditional entropy. More complex prompts can even be advantageous when they specify the semantic distinction more clearly. However, the interpretation of the resulting entropy curve changes. In practice, robustness depends on prompt adherence and on keeping the prompt pair sufficiently controlled. As we discuss in the limitation section and in Appendix C.2, prompt attributes can induce larger distributional shifts, especially in SD 1.5. This makes semantic isolation difficult. Thus, the extension is feasible but best suited to stronger T2I models such as SD XL or SD 3. To demonstrate this feasibility, we have computed the partitioned class-conditional entropy for two pairs of complex prompts using SD XL 1.0 base (https://anonymous.4open.science/r/icml_rebuttal-54F2/figure_sdxl_complex_prompts.pdf).
> These preliminary results suggest that the method can indeed scale to more complex semantic setups and may, in fact, become easier to interpret when the prompt distinction is semantically clearer.
>
> **Response to**:
>
> - *"Could the author extend the empirical results to other diffusion formulations [...]?"*
>
> To address this concern, we reproduced the experiment from Figure 4 using the SD XL 1.0 base, the entropy behaves more smoothly (https://anonymous.4open.science/r/icml_rebuttal-54F2/figure_sdxl_wooden_chair.pdf). In our view, this is encouraging but also unsurprising. The current manuscript already notes that SD 1.5 is limited by prompt adherence and that stronger models should make semantic probing easier. The SD XL results support that interpretation. We have not yet run the same analysis for different formulation so we would prefer to present those as promising directions rather than make an even stronger claim at this stage.
>
> **Response to**:
>
> - *"The conditional entropy estimator in Algorithm 1 relies on a conditional and unconditional noise prediction model. [...] The error of the estimator does not have a rigorous theoretical discussion."*
>
> - *"[...] A detailed analysis of the computational complexity and overhead of the proposed estimator may further strengthen the effectiveness of the proposed estimator."*
>
> - *"Could the author give a detailed analysis of the bias or error of the proposed estimator?"*
>
> We agree that a rigorous error analysis is valuable. The total error decomposes into four contributions: Monte Carlo error (finite sample approximation of Algorithm 1), model approximation bias (posterior approximation and score approximation), null model replacement bias (complement vs. null model approximation), and equal-prior reweighting bias (equal prior vs. true prior). The first term can be controlled by standard Monte Carlo arguments. The second term can partially be analyzed by expressing the posterior update using log-odds. That is, showing that local likelihood ratio errors accumulate additively. From there one can propagate the error through the logistic map and the entropy functional. The remaining two terms are structural approximation terms that we deliberately introduced in the practical implementation. We therefore feel that it is more appropriate to analyze them in practical settings.
>
> However, we also want to be transparent that fully characterizing the model approximation error is genuinely difficult. Our analysis is only valid if the conditional model can reliably separate different conditional distributions. A theoretical bound on this term would require strong assumptions on the score network that are, in our opinion, beyond the scope of this work. To further quantify the remaining error sources, we analyze the convergence of entropy profiles and the effect of online posterior estimation. We conducted ablations on the number of samples required in both GMMs and EDM2-XS (https://anonymous.4open.science/r/icml_rebuttal-54F2/figure_gmm_entropy_estimation.pdf, https://anonymous.4open.science/r/icml_rebuttal-54F2/figure_edm_convergence.pdf), showing convergence with a modest number of trajectories. For GMMs, we further verify that online posterior approximation from the score functions recovers the ground-truth entropy profiles.
>
> On computational complexity: the estimator requires two denoiser evaluations per step per trajectory plus a negligible posterior update, giving $\mathcal{O}(NT)$ overhead, with memory dominated by tracking the posteriors. We will include explicit runtimes in the revision.

---

> > ### Author Rebuttal · Reviewer_j2xx · 2026-04-06
> >
> > I thank the authors for the detailed rebuttal. As my concerns have been adequately addressed, I will update the rating accordingly.

---

> > > ### Author Response · Authors · 2026-04-06
> > >
> > > Thank you for revising your assessment of the paper. We are glad that our rebuttal addressed your concerns.

---

### Official Review · Reviewer_pqHJ · 2026-03-12

**Soundness:** 3
**Presentation:** 3
**Significance:** 3
**Originality:** 2
**Overall Recommendation:** 4
**Confidence:** 4

**Summary:**

This paper *The Entropic Signature of Class Speciation in Diffusion Models* uses partitioned class-conditional entropy, based on ideas from statistical physics, to identify the timing of semantic distinctions in generative diffusion models. By tracking this quantity along the denoising trajectory, reliable signatures of transition regimes are found. Numerical experiments with EDM2-XS (ImageNet-512) and Stable Diffusion 1.5 reveal hierarchical patterns in entropy profiles.

**Compliance With Llm Reviewing Policy:**

Affirmed.

**Final Justification:**

My main concerns regarding the lack of practical utility beyond existing work, as well as the dependence on a computationally intensive grid search, which was scarcely documented in the main text, were satisfactorily addressed in the rebuttal. The authors demonstrate that entropy peaks can serve as class-adaptive guidance schedulers and reduce the number of hyperparameters. As far as I can tell, this partitioned entropy framework is novel and the derivation is elegant. I am raising my rating from 3 to 4.

**Key Questions For Authors:**

1) How can the diagnostic findings be used for the "time-localized control" mentioned in the abstract? Could the entropy production peaks serve as a practical guidance scheduler?
2) What is the specific relationship between the entropy diagnostic and the guidance intervals? To what extent is the analysis based on the expensive grid search, which is only documented in the appendix?
3) Does the addition of attributes degrade the overall image quality? Like the addition of the sleeping cat seems to lead to artifacts at the wall and the chair.

**Limitations:**

yes

**Strengths And Weaknesses:**

*Strengths:*
1) The work is well structured and written. The theory is clearly introduced and motivates the choice of numerical experiments.
2) The application of partitioned entropy to specific semantic distinctions is novel and original, leading to clear and compelling empirical findings on hierarchical commitment. The experiments show clearly visible differences in entropy profiles for different frequency details. Figure 4 and additional examples in the appendix show a strong qualitative signature. It is interesting to see that the blue wall addition in Fig. 4 leads to a visually earlier appearance of the wooden chair. This is supported by the left part of the figure.
3) The derivation of conditional entropy production is elegant and provides an insightful diagnostic.

*Weaknesses:*
1) The presented study is mainly diagnostic and analytical. It is unclear whether the entropy description offers more than an alternative formulation of the same phenomenon that has already been characterized by dynamic and statistical physics (Biroli et al. 2024, Sclocchi et al. 2025). The practical utility is unclear, as the speciation timescale is already known from previous work and the guidance intervals in the ImageNet experiments are based on an expensive grid search (Appendix C). This dependency is hardly documented in the main text, although Figure 3 is entirely based on these grid-searched intervals.
2) There are some problems with the presentation: Figure 1(a) superimposes entropy production curves on a colored density plot with different y-axes and no clear visual separation, making it difficult to evaluate the panel. A color scale for the density plot is missing. Uncertainty bands would be helpful in Figures 1(b), 3, and 4. Lines 12 and 13 must be swapped in Algorithm 1. In line 571, there is an incorrect reference to an equation ("Eq. ??").

---

> ### Author Rebuttal · Authors · 2026-03-31
>
> Thank you for the careful review and constructive suggestions. We have already corrected the missing equation reference in the appendix and the mistake in the pseudocode. In the revision, we will also improve the visual presentation of Figure 1 and add uncertainty bands in the remaining figures, when space allows.
>
> **Response to**:
>
> - *"It is unclear whether the entropy description offers more than an alternative formulation of the same phenomenon that has already been characterized by dynamic and statistical physics (Biroli et al. 2024, Sclocchi et al. 2025). The practical utility is unclear [...]."*
>
> - *"How can the diagnostic findings be used for the "time-localized control" [...]? Could the entropy production peaks serve as a practical guidance scheduler?"*
>
> We agree that prior work has already characterized the existence of a speciation transition and, for variance-preserving processes, its associated logarithmic timescale. Our claim is therefore not that entropy discovers a qualitatively different transition but that it provides a practical and information-theoretic diagnostic of this transition in trained models. In contrast to the instability/eigenvalue perspective from Biroli et al. (2024), entropy can be estimated from a moderate number of generated trajectories without requiring access to the training set and can resolve semantic decisions at the level of individual classes or semantic partitions (https://anonymous.4open.science/r/icml_rebuttal-54F2/figure_edm_convergence.pdf). It therefore turns the previously analyzed transition into a measurable quantity, including under guidance. We will revise the text to make this distinction explicit.
>
> Regarding time-localized control, we have run a follow-up guidance experiment on EDM2-XS showing that class-specific entropy profiles can be used to construct class-adaptive guidance schedules. Concretely, for each class we set one interval bound directly from the entropy profile as the first noise level $\sigma$ at which the estimated partitioned class-conditional entropy reaches zero, and we use a threshold parameter $\theta$ on the profile to determine the class-specific onset of the guidance window. This yields a class-adaptive variant of limited interval guidance (Kynkäänniemi et al., 2024). In contrast to standard LIG, which uses class-independent bounds ($\sigma_{\text{low}}, \sigma_{\text{high}}$), our method shares only a single threshold parameter between classes while allowing the resulting interval onset to adapt to each class-specific entropy trajectory. Our results show that this adaptive scheme matches or slightly improves LIG when optimizing on FID while reducing the number of hyperparameters from 3 to 2 (https://anonymous.4open.science/r/icml_rebuttal-54F2/table_guidance_experiment.pdf). We see this as encouraging evidence that the entropy peak can be used as a practical scheduler, although we would present this as an initial result that requires more research.
>
> **Response to**:
>
> - *"The guidance intervals [...] is based on an expensive grid search. This dependency is hardly documented in the main text [...]. What is the specific relationship between the entropy diagnostic and the guidance intervals? To what extent is the analysis based on the expensive grid search [...]?"*
>
> Indeed, this dependency should be stated more clearly in the main text. In Figure 3 our objective is to provide a diagnostic analysis of a small set of already selected guidance windows in the context of optimality. The entropy diagnostic is intended to explain why selected intervals are effective, not to identify them. In the EDM2 setup, the most effective interval overlaps with the region of highest average entropy production across classes. While preliminary and specific to the current setup, these results are still useful because they show that guidance can be analyzed within the same information-theoretic framework as the unguided dynamics
>
> **Response to**:
>
> - *"Does the addition of attributes degrade the overall image quality? [...]"*
>
> Albeit not the goal of our work to show these effect, we think a plausible explanation is that SD 1.5 inherits caption noise from its training data and is not trained with an explicit decomposition of prompt semantics. Instead, attributes and relations are inferred implicitly from the full prompt, so small attribute edits do not always remain local in the generated image but introduce broader distributional shifts or artifacts. This is also consistent with the SD 1.5 model card, which emphasizes these limitations. In our experience, prompts with richer semantic context are sometimes more stable, but we do not want to overstate this as a general rule. However, the reproduction of Figure 4 using SD XL seems to support this view (https://anonymous.4open.science/r/icml_rebuttal-54F2/figure_sdxl_wooden_chair.pdf).

---

> > ### Author Rebuttal · Reviewer_pqHJ · 2026-04-04
> >
> > The author's response and additional experiment is appreciated - thanks.
> > I will raise my score, as the follow-up experiment demonstrates that entropy peaks can be used to construct class-adaptive guidance schedules, addressing my concern about practical utility.

---

> > > ### Author Response · Authors · 2026-04-04
> > >
> > > Dear reviewer,
> > >
> > > Thank you for the follow-up and for your positive assessment of our additional experiments, we appreciate it.
> > >
> > > We noticed that your current score has not yet been updated and may not reflect your updated evaluation, and wanted to check whether this might have been overlooked.
> > >
> > > Thank you for your time and consideration.

---

### Official Review · Reviewer_LtQY · 2026-03-12

**Soundness:** 3
**Presentation:** 3
**Significance:** 2
**Originality:** 3
**Overall Recommendation:** 3
**Confidence:** 2

**Summary:**

This paper proposes class-conditional entropy and its temporal derivative as operational diagnositcs for semantic emergence in diffusion models. In the GMM analysis, it shows that entropy production is tied to the same symmetry-breaking time scale previously identified in VP diffusion, and it then estimates binary-partitioned entropies in trained models using poeterior tracking, and analyze class-specific entropy windows on EDM2-SX/ImageNet and prompt-attribute-specific windows on Stable Diffusion 1.5, and uses the same framework to analyze how guidance redistributes semantic information over time.

**Compliance With Llm Reviewing Policy:**

Affirmed.

**Final Justification:**

The authors provide additional experimental results but I still feel like there is a gap between the observation and the theoretical assertions and there is little connection further to applications. Therefore, I maintain my score.

**Key Questions For Authors:**

- Under what conditions should the empirical estimator preserve the peak locations of the theoretical $H(\pi(Z)|X_t)$?
- Section 4 argues that VE/EDM does not exhibit the same sharp transition in the rescaled asymptotic analysis, yet the EDM2 experiments still show rather localized entropy-production windows. Is this simply because the asymptotic statement is about a different scaling notion, or it there a depper explanation involving trained-model geometry?
- Appendix C.2 openly acknowledges failure cases and prompt-mismatch issues. More detail on robustness would help readers understand whether the method is already usable as a practical probe, or mostly a proof of concept on carefully chosen prompt pairs.

**Strengths And Weaknesses:**

### Strengths
- The paper connects the statistical-physics symmetry breaking viewpoint to an information-theoretic quantitiy that is operationally measurable in trained models.
- The EDM2/ImageNet and Stable Diffusion prompt-partition experiments are visually and conceptually compelling.

### Weaknesses
- The sign and prefactor in the entropy-production formula are not presented consistenly between the main text and appendix, making it harder to verify what quantity is actually being analyzed.
- In practice, the paper replaces the class complement by the unconditional model,fixes the partition prior to 0.5, and under guidance changes the expectation to a guided cross-entropy-like quantity. These approximations are reasonable but weaken the clean theory to empirical connection.

---

> ### Author Rebuttal · Authors · 2026-03-31
>
> We thank the reviewer for the constructive feedback. We will update the manuscript to correct the noted inconsistencies. Below, we address the remaining concerns in turn.
>
> **Response to:**
>
> - *"[...] theory to empirical connection."*
>
> Indeed, these practical choices weaken the exact correspondence between the clean theoretical quantity and the empirical estimator, and we will make that distinction clearer in the revision. Our claim is therefore not that the entropy values are exactly preserved, but that the estimator still localizes the same semantically meaningful transition window.
>
> For our binary probes, the relevant quantity is the partitioned class-conditional entropy in Eq. (17), which is governed by a single log-odds variable,
>
> $\Lambda(x,t):=\log \frac{\gamma_1(x,t)}{\gamma_0(x,t)}.$
>
> Changing the partition prior only adds a constant shift,
>
> $\Lambda(x,t)=\log \frac{p(X_t=x\mid \pi=1)}{p(X_t=x\mid \pi=0)}+\log \frac{p(\pi=1)}{p(\pi=0)}.$
>
> Thus, replacing the true prior by $p(\pi)=0.5$ does not affect the time-dependent likelihood-ratio term, and so should not change the leading speciation timescale, only shift the peak within the same critical window. More generally, in many settings we do not even know what the correct priors should be—for example, for prompt-defined semantic partitions—so the balanced choice is also a practical and well-defined convention. Empirically, using the true ImageNet priors does not qualitatively change the entropy curves, it mainly shifts them slightly to earlier times and increases estimator variance (https://anonymous.4open.science/r/icml_rebuttal-54F2/figure_edm_prior_analysis.pdf).
>
> For class-vs-class comparisons, this balanced prior feels natural. For class-vs-complement, it turns the estimator into a balanced overlap probe, which, as noted in Appendix B, is equivalent to a Jensen--Shannon divergence.
>
> The same logic applies when replacing the exact class complement with the unconditional model. This approximation is not exact, and we will make that clearer, but the estimator remains a useful proxy for identifying semantically meaningful transition intervals. In the text-conditioned setting, the probes instead use binary prompt partitions targeting specific semantic edits. We will also analyze, in the Gaussian mixture setting, how this approximation affects the estimator as the number of classes increases.
>
> **Response to:**
>
> - *"[...] EDM does not exhibit the same sharp transition [...]"*
>
> Our asymptotic result for VE/EDM is not intended to suggest that localized transition windows cannot appear, but only that, unlike in the VP case, they do not become progressively narrower in normalized time as the dimension grows. In other words, our results are fully consistent with the presence of localized entropy-production transitions in the EDM case as well. Indeed, our plots illustrate this behavior precisely: the EDM peaks are clearly localized, but they do not sharpen under normalized time. We realize this point may have been obscured by the presentation of our plots in normalized time. In standard time coordinates, the EDM transition window broadens with dimension (Figure 2(b)), whereas in normalized time (as in our empirical results) it remains of comparable width rather than shrinking further, as happens in the VP case. If the reviewer believes this would improve clarity, we would be happy to add a supplementary plot, analogous to Figure 2, showing the EDM entropy curves in normalized time.
>
> As noted in the paper, the practical significance of this asymptotic distinction remains unclear, but we believe it is worth reporting since it does not seem to be widely discussed in the literature.
>
> **Response to:**
>
> - *"[...] More detail on robustness [...]"*
>
> We have already run additional robustness checks for SD 1.5 and extended the experimental setting to SD XL (https://anonymous.4open.science/r/icml_rebuttal-54F2/figure_sdxl_wooden_chair.pdf) where entropy profiles are smoother and samples align better with prompts. However, we do not want to overstate the present robustness of the method. As discussed in the limitations section, there are two main failure modes in SD 1.5. First, the model may simply fail to realize the intended attribute (for example, the added object is absent). Secondly, the variant prompt can induce a distributional shift relative to the base prompt, in which case the estimated entropy no longer isolates the targeted semantic factor. The entropy profile still remains a valid overlap diagnostic for the two prompt conditioned distributions, but its interpretation as an attribute probe then becomes weaker. To make this clearer for readers, we will add a short paragraph on robustness summarizing the successful replications, the observed failure modes, and the fact that stronger models such as SD XL already behave more consistently.

---

> > ### Author Rebuttal · Reviewer_LtQY · 2026-04-04
> >
> > I thank the authors for providing additional experimental results.

---

> > > ### Author Response · Authors · 2026-04-04
> > >
> > > Dear reviewer,
> > >
> > > Thank you for the follow-up response and for taking the time to review our additional experiments, we’re glad the clarifications addressed your concerns.
> > >
> > > We noticed that your current score has not yet been updated and may not reflect your updated assessment, and wanted to check whether you would like to revise it.
> > >
> > > We appreciate your time and consideration.

---

### Official Review · Reviewer_pc1s · 2026-03-19

**Soundness:** 3
**Presentation:** 3
**Significance:** 2
**Originality:** 3
**Overall Recommendation:** 4
**Confidence:** 3

**Summary:**

Starting from the mixture of Gaussian distribution, this paper proposes class-conditional entropy and its entropy production rate as operational tools for tracking semantic emergence during the generative process of diffusion models. From the theoretical perspective, this work shows that entropy production concentrates on the same logarithmic time scale as the speciation time for VP-based models, as previous work predict. For the VE-EDM, similar phenomenon does not exhibit a sharp phase.  After that, this works further introduce "partitioned class-conditional entropy" to resolve semantic branching at different levels of abstraction. Experiments on EDM2-XS and Stable Diffusion 1.5 validate the theoretical predictions and demonstrate how classifier-free guidance redistributes information flow across the denoising trajectory.

**Compliance With Llm Reviewing Policy:**

Affirmed.

**Final Justification:**

As the additional experiments on SD and additional theoretical guarnatee address my concerns, I will maintain my positive score.

**Key Questions For Authors:**

Please see the  Weakness.

1. Can you provide a calibration experiment on a non-Gaussian synthetic distribution where ground truth conditional entropy is computable (e.g., a 2D mixture of non-isotropic Gaussians, or a simple multi-modal distribution trained with a small diffusion model).

**Limitations:**

yes

**Strengths And Weaknesses:**

Strength:

1.	The paper aims to connect the existing symmetry-breaking/speciation theoretical picture to an information-theoretic quantity (class-conditional entropy) that can be directly estimated in trained models. This fills an important gap between theoretical analysis and practical diagnostics.

2.	The proposed method can be directly applied to pretrained models without modifying architectures or retraining, which is useful in applications.

3.	The comparison between VP and EDM is interesting and valuable.

Weakness:

1.	The theoretical analysis is entirely based on equiprobable Gaussian mixtures with equidistant means and isotropic covariance. I understand that this is a meaningful step to analyze this setting. I wonder if it is possible to extend the analysis to the low-dimensional data or a more general GMM.

2.	Algorithm 1 is the central methodological contribution enabling all experimental results, yet its accuracy is never rigorously validated. More specifically, it would be better to conduct experiment son a non-Gaussian synthetic distribution (e.g., 2D/3D mixture of non-isotropic components) and comparing Algorithm 1's output against numerical integration.
3.	It would be better to conduct quantitative convergence study.How does the estimated entropy change as a function of sample size $N$? What is the variance across different random seeds?

---

> ### Author Rebuttal · Authors · 2026-03-31
>
> Thank you for the constructive feedback.
>
> We agree that the original submission would benefit from a broader theoretical treatment and an explicit validation of Algorithm 1. In the revision, we will therefore add two sets of new results. Firstly, we will replace our current theoretical framework with a substantially more general analysis. We rewrite the class-conditional entropy production in a pairwise form:
>
> $\dot{H}[Z\mid X_t]=\frac{g_t^2}{4}\sum_{r,s=1}^N\mathcal E_{r\to s}(t),$ with
>
> $\mathcal E_{r\to s}(t)=w_r\mathbb E_{X_t\sim p_t(\cdot\mid Z=r)}\left[\pi_s(X_t,t) ||s_r(X_t,t)-s_s(X_t,t)||^2\right],$ and
>
> $\pi_r(x,t)=\mathbb P(Z=r\mid X_t=x)$ and $s_r(x,t)=\nabla_x\log p_t(x\mid Z=r)$.
>
> Writing the posterior through the log-odds
>
> $\Lambda_{rs}(x,t)=\log\frac{w_r}{w_s}+\log P_r(x;t)-\log P_s(x;t),$
>
> we obtain
>
> $\pi_s(X_t,t)=\frac{e^{-\Lambda_{rs}(X_t,t)}}{1+\sum_{u\neq r}e^{-\Lambda_{ru}(X_t,t)}}.$
>
> Hence, $\mathcal E_{r\to s}(t)$ is appreciable only when $\Lambda_{rs}(X_t,t)=O(1)$. For $\Lambda_{rs}\gg 1$, the posterior factor is exponentially suppressed, whereas after equilibration, the score difference $||s_r-s_s||^2$ becomes small. We then connect this entropy-production picture to the recent paper on the general theory of speciation transitions [1]. In that framework, for samples generated from class $r$, the log-odds admit the large-dimensional decomposition
>
> $\Lambda_{rs}(X_t,t)=\log\frac{w_r}{w_s}+d\big(f_{rr}(t)-f_{rs}(t)\big)+\big(\delta f_{rr}(X_t,t)-\delta f_{rs}(X_t,t)\big),$
>
> where $f_{rr}(t)-f_{rs}(t)$ is the average free-entropy gap and the last term captures its fluctuations. This plays the same role as in our Gaussian analysis: the average gap gives the deterministic class advantage, while the fluctuation term determines the spread of the log-odds. The difference is, in the general case, the fluctuations need not be Gaussian, so one studies them through a large-$t$ expansion in powers of $e^{-t}$. Speciation occurs when the mean gap becomes comparable to its fluctuations. Before speciation, the posterior competition term is exponentially suppressed, while sufficiently after speciation, the score fields have already merged, so the contribution vanishes again. Importantly, this extension covers covariance-driven, not only mean-driven, separation. In the large-$t$ regime,
>
> $\lambda_{rs}(t)=\begin{cases}e^{-2t}A_{rs} & A_{rs}>0, \ e^{-4t}B_{rs} & A_{rs}=0,\end{cases}$
>
> where $A_{rs}$ and $B_{rs}$ quantify mean and covariance separation. The speciation window is defined by $\lambda_{rs}(t)=O(1)$. In the Gaussian case, this gives $t_s\sim \frac{1}{2}\log d$ for mean-separated classes and $t_s\sim \frac{1}{4}\log d$ for covariance-driven separation. More generally, different class pairs may have different $\lambda_{rs}$ and thus different speciation times, naturally yielding multiple entropy-production peaks in hierarchical mixtures.
>
> To complement the theory, we add two experiments that go beyond the original setup (https://anonymous.4open.science/r/icml_rebuttal-54F2/figure_general_setting_entropy.pdf): (i) two concentric Gaussians, where the classes have identical means but distinct covariances, and (ii) a hierarchical four-Gaussian mixture with both covariance- and mean-driven mergers. These examples are useful since they directly probe settings that are not captured by the original isotropic mean-separated analysis, and they exhibit the multi-stage entropy transitions predicted by the extended theory. At the same time, we keep the original Gaussian analysis in the paper since it remains the cleanest solvable example for explaining the underlying mechanism. It also makes transparent the contrast between VP and EDM SDEs, and we agree that this distinction is an interesting and valuable contribution.
>
> Second, since Algorithm 1 is central to all empirical results, we will add the suggested calibration study on Gaussian mixtures. That is, how the estimated class-conditional and partitioned class-conditional entropy estimates compare to the ground truth as we increase the number of samples (https://anonymous.4open.science/r/icml_rebuttal-54F2/figure_gmm_entropy_estimation.pdf). Additionally, we will add a convergence analysis conducted on the ImageNet/EDM2 setup for each of the three example classes in Figure 1 where we show how the entropy estimate varies with sample size and for different random seeds (https://anonymous.4open.science/r/icml_rebuttal-54F2/figure_edm_convergence.pdf).
>
> We hope these additions address both concerns. The revised theory is no longer restricted to the original Gaussian setting and, more importantly, demonstrates that class-conditional entropy can detect multiple forms of speciation in the data. In addition, the estimator is now supported by direct calibration and convergence studies.
>
> [1] Achilli, Beatrice, et al. *Theory of Speciation Transitions in Diffusion Models with General Class Structure.* arXiv:2602.04404

---

> > ### Author Rebuttal · Reviewer_pc1s · 2026-04-03
> >
> > I thank the authors for addressing my theoretical and experimental concerns.
> >
> > * The generalization to pairwise log-odds with general class structure addresses my main concern.
> > * Furthermor, the calibration and convergence studies address W2 and W3.
> > * I also happy for the additional experiments on SDXL. I also understand the other reviewers concerns on SoTA T2I models such as FLUX.1-dev. Howver, I think it is a good start from SDXL
> >
> > Hence, I will maintain my positive score 4.

---

> > > ### Author Response · Authors · 2026-04-06
> > >
> > > Thank you very much for your positive assessment of our revisions. We appreciate your careful reading and are glad that our changes addressed your concerns.

---

### Decision · Program_Chairs · 2026-04-30

**Decision:**

Accept (regular)

**Comment:**

This paper proposes an information-theoretic diagnostic for semantic emergence in diffusion models via class-conditional entropy and its partitioned variants. The reviewers agree that the paper is well motivated, technically interesting, and offers a novel operational perspective on speciation phenomena previously studied mainly through dynamical or statistical-physics analyses.

The main concerns were about the limited generality of the original theory, the lack of validation and error analysis for the estimator, and the still somewhat preliminary evidence for practical utility. The authors’ rebuttal addressed many of these points convincingly by broadening the theoretical framing, adding calibration and convergence experiments, clarifying the approximation gap between theory and practice, and providing additional evidence that entropy profiles can inform adaptive guidance schedules. While one reviewer remains unconvinced about the strength of the theory-to-practice connection and downstream applicability, the overall post-rebuttal picture is positive, with three reviewers at weak accept and multiple concerns explicitly marked as resolved.

Overall, this as a solid and novel contribution with moderate impact and some remaining limitations in scope and validation, but with enough strength to merit inclusion in the program.